# A tectonic carpet of Variscan flysch at the base of a rootless accretionary prism in NW Iberia: U-Pb zircon age constrains from sediments and volcanic olistoliths

Emilio González Clavijo[1], Ícaro Dias da Silva[2,3], José R. Martínez Catalán[4], Juan Gómez Barreiro[4], Gabriel Gutiérrez-Alonso[4], Alejandro Díez Montes[1], Mandy Hofmann[5], Andreas Gärtner[5], Ulf Linnemann[5]

[1]Instituto Geológico y Minero de España. Plaza de la Constitución 1, 3º, 37001 Salamanca, Spain
[2]Instituto Dom Luiz, Faculdade de Ciências, Universidade de Lisboa, Campo Grande, 1749-016 Lisboa, Portugal
[3]Departamento de Geologia, Faculdade de Ciências, Universidade de Lisboa, Campo Grande, Lisboa, 1749-016, Portugal;
[4]Departamento de Geología, Universidad de Salamanca. Plaza de la Merced, s/n, 37008 Salamanca, Spain
[5]Senckenberg Naturhistorische Sammlungen Dresden, Königsbrücker Landstr. 159, D-01109 Dresden, Germany

*Correspondence to*: Ícaro Dias da Silva (ipicaparopo@gmail.com)

**Abstract.** The allochthonous complexes of Galicia – Trás-os-Montes Zone (NW Iberia) are part of a rootless tectonic stack
which preserves part of a Variscan accretionary prism. They are formed by individual tectonic slices marked by specific tectono-metamorphic evolutions, which were piled up in a piggy-back mode onto its relative autochthon, the Central Iberian Zone (CIZ). Allochthony decreases from the structurally upper thrust sheets towards the lower ones. The lowermost unit of the stack is known as the Parautochthon or Schistose Domain. It is characterized by low metamorphic grade in contrast with higher temperatures and/or pressures estimated for the overlying allochthonous units, and shares the stratigraphic sequence
with the underlying autochthon. The Parautochthon is divided in two structural and stratigraphic sub-units: i) the Lower (LPa) is made of synorogenic flysch-type sediments with varied turbiditic units and olistostrome bodies, showing Upper Devonian-lower Carboniferous age according to the youngest zircon populations and fossiliferous content; ii) the Upper (UPa) is composed of highly deformed preorogenic upper Cambrian-Silurian volcano-sedimentary sequence comparable with the nearby autochthon and to some extent, also with the high-P and low-T Lower Allochthon laying structurally above. The UPa
was emplaced onto the LPa along the Main-Trás-os-Montes Thrust, and the LPa became detached from the CIZ relative autochthon by a regional-scale structure, the Basal Lower Parautochthon Detachment, which follows a weak horizon of Silurian carbonaceous slates.

A review on the detrital zircon studies on the synorogenic LPa complemented by zircon dating of 17 new samples is presented here. The results support the extension of the LPa underneath the NW Iberia allochthonous complexes, from Cabo Ortegal, to
Bragança and Morais Massifs. Its current exposure follows the lowermost tectonic boundary between the Galicia – Trás-os-Montes (allochthon) and Central Iberian (autochthon) zones. The youngest zircon age populations' point to a maximum sedimentation age for the LPa formations ranging from Famennian to Serpukhovian and supports the piggy-back mode of emplacement of the Galicia – Trás-os-Montes Zone, of which it represents the latest imbricate.

The zircon age populations in the LPa allow constraining the sedimentary provenance areas, showing the intervention of nearby
sources (mostly the UPa) and/or multiply recycled/long transport sediments with typically N-Central Gondwana age
fingerprint, also found in the Lower Allochthon, UPa and Autochthon. Complementary geochronology of volcanic olistoliths
trapped in the LPa sediments and of late Cambrian to Upper Ordovician rhyolites from the UPa is also presented. It shows a
direct relationship between the major blocks source area (UPa) and the setting place (LPa). Old zircon age patterns show that
the LPa sedimentary rocks were recycled from detrital rocks of the allochthon (advancing wedge) and the nearby autochthon
(peripheral bulge).

## 1 Introduction

Synorogenic marine basins encompass most of the known marine geodynamic settings, from active to passive earthquake-
prone margins (Dickinson and Valloni, 1980; Garzanti et al., 2007; DeCelles, 2012). They are found associated with Archean
to Phanerozoic orogens (e.g. Liang and Li, 2005; Wilmsen et al., 2009; Mulder et al., 2017; Kusky et al., 2020) and hold key-
evidences to understand the geographic and geodynamic evolution of modern and ancient orogenic belts.

A common sedimentary feature of all synorogenic marine basins is the presence of earthquake-triggered turbiditic flows that
promote a variety of sedimentary facies, from cohesive rhythmic flysch sequences to chaotic large-scale mass-wasting bearing
heterometric size blocks or olistoliths (Franke and Engel, 1986; Coleman and Prior, 1988; Eyles, 1990, Festa et al., 2020), also
denominated Block-in-Matrix formations (hereafter referred as BIMF) (Festa et al. 2016). Because the most probable source-
areas of the sediments and olistoliths that fed these basins are in the surrounding orogenically active highs, their stratigraphy
can provide important clues on the orogen relief variation along space and time (e.g. Ducassou et al., 2014; Chiocci and
Casalbore 2017).

The synorogenic basins formed during continental convergence are gradually incorporated into the active orogenic edifice as
tectonic slices in the accretionary complex, especially at the base, forming a tectonic carpet (Festa et al., 2019; Kusky et al.
2020, and references therein). The progradation of the tectonic front and the basin depocenter, favors a systematic intrabasinal
sedimentary recycling (i.a. wild flysch) and mixing of synorogenic sediments with other external sources (e.g. Franke and
Engel, 1986; Bütler et al., 2011) producing mixed signals of not straightforward paleogeographic interpretation. The key to
address this problem rests in the regional study of the basin stratigraphy, including the flysch sequences, the mass-wasting
deposits, and the petrography of the olistoliths (Festa et al., 2019, 2020). Complementary, the detailed geological recognition
of the basement and surrounding areas of the synorogenic basins is crucial to identify discriminatory aspects that can help to
constrain different variables, such as possible sources, sediment transport distance, regional and local tectonic settings, and
paleogeographic limitations (e.g. Alonso et al., 2015; Festa et al. 2016, Krastel et al., 2019).

The sedimentological models can be refined using detrital zircon geochronology. This tool is commonly used to trace source-
to-sink relationships, through different statistical approaches that compare the zircon age populations present in detrital rocks
(e.g. Meinhold et al., 2011, 2013; Linnemann et al., 2012, 2014). One of the most rigorous procedures for zircon age

populations similarity/dissimilarity analysis is the Multidimensional Scaling (MDS) (Vermeech, 2018), that provides a graphic output of the Kolmogorov-Smirnov test when using a large number of samples with individual zircon age populations (Gutiérrez Alonso et al., 2020; Pereira et al., 2020a).

In the southwestern edge of the European Variscan Belt (Fig. 1), the Iberian Massif preserves some of the best examples of
Phanerozoic synorogenic marine basins, that reflect different tectonic settings along the belt during the Upper Devonian-late Carboniferous collision of Laurussia and Gondwana to from Pangea (e.g. Pereira et al., 2012, 2017; Oliveira et al., 2019b). In SW Iberia, the Late Devonian – late Carboniferous flysch basins appear on both sides of the oceanic suture that separates Laurussia from Gondwana (Silva et al., 1990; Braid et al., 2011; Pereira et al., 2012; Pérez-Cáceres et al., 2017). On the Gondwana-side, the Late Devonian – early Carboniferous marine sedimentation had Gondwana-type sources with massive
contribution of intrabasinal volcanism in the Tournaisian-Visean period (Pereira et al., 2012, 2020a). On the Laurussian side, sediments that filled the synorogenic marine basins resulted from intrabasinal recycling processes and source areas located on both continents. In this case, the sediments were systematically imbricated at the base of the advancing orogenic front, towards inland Laurussia from the Late Devonian (Pulo do Lobo Zone) to the upper Carboniferous (southwestern South Portuguese Zone) (Pereira et al., 2012, 2020a; Pérez-Cáceres et al., 2017; Braid et al., 2011; Jorge et al., 2013, Rodrigues et al., 2015).

This study presents new data and analysis of synorogenic rocks from marine basins located in the hinterland (internal zones) of the Variscan belt in NW Iberia. While synorogenic basins in the external zones of the orogen (the foreland fold and thrust belt) have been classically classified as foreland basins laying on top of N-Gondwana Cambrian to Upper Devonian passive margin sequences (Marcos and Pulgar, 1982; Pastor-Galán et al., 2013; Gutiérrez-Alonso et al., 2015), the studied basins in this work are located in the hinterland and were deposited over rocks showing pervasive strain and that were metamorphosed
to different degrees (Martínez Catalán et al., 2004, 2008, 2016; Dias da Silva et al. 2015). The sedimentary sources of the synorogenic flysch and BIMF deposits are related to the development and unrooting of a Variscan accretionary prism (the Galicia-Trás-os-Montes Zone) onto Gondwana, and to the development of a peripheral bulge affecting the extensive passive margin of Gondwana (e.g. González Clavijo and Martínez Catalán, 2002; Keller et al., 2008; Dias da Silva et al. 2015). Some of the Variscan hinterland synorogenic basins have been incorporated into the base of the allochthonous wedge as a
parautochthonous unit, and then emplaced onto the autochthonous terrain of NW Iberia (Dias da Silva et al. 2015, González Clavijo et al., 2016). The basin depocenter migrated towards inland Gondwana from the Late Devonian to the late Carboniferous following the progression of the orogenic front, from the Galicia-Trás-os-Montes Zone (GTMZ), to the Central Iberian Zone (CIZ), where synorogenic deposits are preserved in the San Clodio Series (Martínez Catalán et al., 2016), possibly following to the West Asturian-Leonese zone (WALZ), although no synorogenic deposits linked to this basin are preserved on
it, and finally to the foreland Cantabrian Zone (CZ) (Merino-Tomé et al., 2017; Gutiérrez-Alonso et al., 2020).

In both NW and SW Iberia synorogenic basins, zircon geochronology has been used to constrain sedimentary provenance based on the fingerprint of sources and basin stratigraphic units (Braid et al., 2011; Pereira et al., 2012; 2014; Jorge et al., 2013; Pastor-Galán et al., 2013; Rodrigues et al., 2015; Martínez Catalán et al., 2016; Pérez-Cáceres et al., 2017). While in the SW and in the CZ recent works have demonstrated the importance of the MDS statistical approach in identify the relationships

between source and sink (Gutiérrez-Alonso et al., 2020; Pereira et al., 2020b), in NW Iberia variscan hinterland this approach has not been applied up to date.

In this work we present new field observations that revise previous interpretations, supported by new structural data and U-Pb geochronology of igneous and detrital zircon grains, which enables a new vision of the parautochthonous units of the GTMZ here referred as Upper Parautochthon (UPa; preorogenic) and Lower Parautochthon (LPa; synorogenic) (Dias da Silva et al.,

2015). Our findings support the extension of these units to different sectors of the GTMZ, extending the area covered by the UPa and LPa to virtually below all the allochthonous complexes of NW Iberia. Field work has revealed the diverse types of flysch complexes and mélanges present in the LPa, commonly obscured by Variscan polyphasic and pervasive deformation. Tectonic and sedimentary mélanges have been recognized; they combine to produce polygenetic mélanges using the terminology by Festa et al. (2019; 2020). Moreover, the detrital zircon age fingerprinting offers a new general view of the LPa

geotectonic setting at Variscan times and sets new constrains on the possible source-areas of the synorogenic sediments, olistoliths and blocks. We consider that this review leds to a better understanding on the paleogeography and geodynamic setting of the Late Devonian-lower Carboniferous flysch basins in NW Iberia and its incorporation into a general model for the synorogenic basins of the Iberian Variscan Massif.

## 2 Geological setting

Within the rootless Variscan accretionary wedge which forms the Variscan Massif of NW Iberia – the so called GTMZ – (Figs. 2 and 3) (Ribeiro, 1974, Schermmeron and Kotsch, 1984, Martínez Catalán et al. 2009; Ballèvre et al., 2014; Martínez Catalán et al. 2014; Azor et al., 2019) a significantly distinct structural unit has been identified and interpreted as a remnant of former oceanic realms (Lower Ordovician Rheic Ocean and an Early Devonian suprasubduction ophiolite). This tectonic unit marks the putative suture zone of the Laurussia-Gondwana continental collision that partially led to the formation of Pangea; this unit

is known as Middle Allochthon (MA) or Ophiolitic Unit (Gómez Barreiro et al., 2007; Martínez Catalán et al., 2009; Stampfli et al., 2013; Ballèvre et al., 2014; Arenas and Sánchez-Martínez, 2015; Azor et al., 2019) and currently separates the so-called Upper Allochthon (Upper units) and the Lower Allochthon (Basal units). The Upper Allochthon (UA) is considered a far-travelled ribbon-shaped terrane that drifted away in the Lower Ordovician from the Gondwanan margin during the opening of the Rheic Ocean, and accreted to Laurussia in the Silurian (Gómez-Barreiro et al., 2007). It includes two tectonically stacked

units, presenting early-Variscan (c. 390-380 Ma) HP/HT and Cambro-Ordovician IP/M to HT metamorphism respectively (Martínez Catalán et al., 2019). The Lower Allochthon (LA) is made of a set of nappe folds and tectonic slices (Farias et al., 1987; Díez Fernández et al., 2010; Dias da Silva et al., 2014; 2015) considered to represent the most seaward rim of continental Gondwana (Murphy et al., 2008) which underwent continental subduction (HP/L to MT metamorphism) and obduction (retrogression to amphibolite and greenschist facies) recording the inception of Variscan continental collision at ca. 370-360

Ma (Munhá et al., 1984; Gil Ibarguchi and Dallmeyer, 1991; Arenas et al., 1995, 1997; Gil Ibarguchi, 1995; Rubio Pascual et al., 2002; Rodríguez et al., 2003; López-Carmona et al., 2010, 2014). This latter unit, (LA) despite being interpreted as part of

the Gondwanan passive margin ensemble and, being part of the lower plate in the collisional edifice, is classically considered as part of the allochthonous realm in the region but not belonging to the exotic terraines (Ribeiro, 2013; Ribeiro and Sanderson, 1996; Ribeiro et al., 2007). Structurally below the LA, a tectonic unit, displaying low metamorphic grade, separates the above-mentioned allochthons from their relative autochthon, the Central Iberian Zone (CIZ). This unit, named Schistose Domain (Farias et al., 1987) or Parautochthon (Pa) (Ribeiro et al., 1990; Martínez Catalán et al., 1997) was considered as a thick Silurian sequence, lacking correlation with the condensed Silurian graphite-rich sequences of the underlying autochthon (CIZ). Nevertheless, the paleogeographic affinity of both domains was highlighted by identical N-Gondwana Silurian graptolite and conodont faunas (Sarmiento et al. 1998; Piçarra et al., 2003; 2006a; 2006b). However, the stratigraphy and structural features of the lowermost tectonic sheets of the Pa, directly above the CIZ, were interpreted as a synorogenic basin with possible (Middle-Late) Devonian age (Antona and Martínez Catalán, 1990; González Clavijo and Martínez Catalán, 2002; Martínez Catalán et al., 2004; Pereira et al., 2009; Rodrigues et al., 2013).

The attempt to better understand the tectonostratigraphy of the Pa, led to a later division in two tectonically stacked units, Upper and Lower Parautochthon (UPa and LPa), in the sense firstly proposed by Rodrigues et al. (2006; 2013) and updated by Dias da Silva et al. (2014; 2015; 2016). This division limits the UPa to a pre-Variscan upper Cambrian-Silurian sequence comparable with the CIZ and LA that was affected by Variscan recumbent folds and thrusts; and defines the LPa as an imbricated thrust sequence bearing slices of a foreland synorogenic basin, with the younger slices in the transition to the CIZ (Martínez Catalán et al., 2016). The thrust fault structures bounding the lower tectonic sheets of the GTMZ are (Figs. 2 and 3):

i)     the LA Basal Thrust (LABT, Figs. 3 and 4) or basal thrust of the Centro-Transmontano thrust complex (in the meaning of Ribeiro et al. 1990) represents the roofing thrust of the Parautochthon;

ii)    the UPa-LPa thrust system (Main Trás-os-Montes Thrust, MTMT; Ribeiro, 1974; Ribeiro and Ribeiro, 2004; Meireles et al., 2006; Pereira et al., 2006) is an up to 1000 m thick gently dipping low-grade shear-zone interpreted to be caused by the thrusting of the upper Cambrian-Silurian (preorogenic UPa) sequence onto the synorogenic LPa, producing significant crustal thickening during the Tournaisian-Visean stage (Dias da Silva et al., 2014; 2015; 2016; 2020; Azor et al., 2019);

iii)   At the base of the LPa another major bedding-parallel fault named the Basal Lower Parautochthon Detachment (BLPD), also gently dipping, is the sole fault separating the synorogenic imbricated slices from the structurally underlying non imbricated autochthon (Dias da Silva et al., 2014). The BLPD was developed using a slip-favorable stratigraphic unit, the autochthonous Silurian carbonaceous-siliceous slates (SCSS) (González Clavijo and Martínez Catalán, 2002; Dias da Silva et al., 2014).

The northern CIZ autochthonous domain consist of an Ediacaran to Lower Devonian preorogenic sequence, disturbed by two regional unconformities, and including c. 490 Ma to 460 Ma felsic to intermediate, locally mafic magmatism, Floian Armorican-type quartzites, a Middle-Upper Ordovician mostly detrital sequence and the SCSS (e.g. Sousa, 1984; Valladares

et al., 2000; Gutiérrez- Marco et al., 2019; Sánchez García et al., 2019). In the autochthon laying immediately below the BLPD, tectonic overburden mainly occurred due to the action of thin-skinned imbricated thrust-duplexes rooted in the BLPD, developed in the LPa tectonic units (Fig. 2 and sections 4 and 5 in Fig. 3) (Dias da Silva et al., 2020). In the sectors where the LPa is present, thickening in both CIZ and LPa was rapidly attenuated by the succeeding synorogenic extensional processes (Dias da Silva et al., 2020).

In the studied area, the UPa and CIZ underwent regional Barrovian metamorphism ($M_1$) through the early Variscan compressive events ($C_1+C_2$ on the Martínez-Catalán et al., 2014 proposal; c. 360-330 Ma) which were followed by a complex extensional ($E_1$-$M_2$; c. 340-320 Ma) and compressional ($C_3$-$M_3$; c. 318-300 Ma) tectonothermal history (Dallmeyer et al. 1997, Azor et al., 2019; Dias da Silva et al., 2020, and references therein).

The LPa synorogenic ensemble was also affected by metamorphism of very low- to low-grade (chlorite zone). Attempts to
discriminate if the metamorphic grade was lower in the synorogenic units than in the preorogenic units using illite crystallinity (Antona and Martínez Catalán, 1990) and Colour Alteration Index in conodonts (Sarmiento and García-López, 1996; Sarmiento et al., 1997) were inconclusive. Petrographic observations made by Matte (1968) in the synorogenic deposits of the San Clodio in the CIZ to the SE of Monforte (Fig. 2) and in the underlying Ordovician sequence, and by Dias da Silva et al. (2020) in the LPa and CIZ in the eastern rim of Morais Complex, shows similar low grade epizone metamorphism in both pre-
and synorogenic sequences. However, the San Clodio flysch rests unconformably above the reverse limb of a large $C_1$ recumbent syncline whose axial planar cleavage is more evolved than that of the flysch above (Martínez Catalán et al., 2016). And it is also older: c. 360 Ma (Dallmeyer et al., 1997), while detrital zircons are as young as 324 Ma in the San Clodio Series and 340 Ma in the synorogenic deposits of Trás-os-Montes (Martínez Catalán et al., 2004, 2008, 2016), age of emplacement of the Allochthonous Complexes during $C_2$. The first foliation in the preorogenic metasediments of the UPa and CIZ is axial
planar to recumbent folds of the $C_1$ event, and predates the main foliation in the synorogenic deposits. But a second, low grade penetrative foliation was developed in the UPa, LPa and CIZ during the emplacement of the Allochthon ($C_2$). This second regional foliation is the one showing similar aspect and metamorphic conditions in both UPa and LPa ensembles, but in the latter it represents the first tectonic fabric.

**3 Review of the synorogenic marine sequences in NW Iberia variscan hinterland**

The internal zones of the orogenic belts are considered areas with scarcely preserved related synorogenic sequences because of the subsequent denudation caused by the orogenic relief (Martínez Catalán et al., 2008), but they might be preserved in the core of synclines or below post-depositional thrust. In other parts of the Variscan belt, Franke and Engel (1986) described tectonic slices carrying synorogenic sedimentary units from more internal areas. In NW Iberia, while there is a complete record of the synorogenic deposits in the foreland fold and thrust belt (CZ. e.g. Marcos and Pulgar, 1984; Merino-Tomé et al., 2017),
in the deeply eroded internal part of the chain, the firstly identified synorogenic sequence was the San Clodio Series (Matte, 1968), which is preserved at the core of the late-Variscan Sil Syncline in northern Iberia (Figs. 2, 4 and 5, and cross section 2

in Fig. 3). It consists of a rhythmic turbiditic sequence of pelite and greywacke (Riemer, 1966; Pérez-Estaún, 1974) including Upper Devonian or even older fossil plant fragments (Pérez-Estaún, 1974). Previous detrital zircon studies (samples SO-1 and SO-2; location of all samples from previous studies are ploted in the map of figure SF1.1 of the Supplementary File) have reinforced the synorogenic character and supported an Upper Mississippian maximum depositional age (Martínez Catalán et al., 2004) according to the youngest ages found in the detrital zircon population of c. 324 Ma. Towards the base it displays exotic lithic blocks and pebbles, that is, extrabasinal rocks derived from the basement exposed in the basin surrounding highs. These include carbonaceous chert, quartzite, slate, gneiss and granite. Intrabasinal ("native") lithified intraclasts and soft pebbles also occur (Riemer, 1966). The San Clodio Series is often separated from the underlying Ordovician formations by a few metres of Silurian black shales (SCSS) which were mylonitized forming a basal detachment (Barrera Morate et al., 1989). However, the unconformity was preserved from reactivation at a few places (Martínez Catalán et al., 2016). Following Festa et al. (2019; 2020) terminology, the San Clodio Series represents a coherent primary succession above a sedimentary block-in-matrix (olistostrome).

The presence of a variably deformed SCSS unit below the synorogenic lithostratigraphic units in the LPa, as described for the Sil Syncline, is a constant feature in all the areas incorporated to this study. The BLPD is a first-order structure that forms a complex arrangement of stacked tectonic slices depicting diverse paterns. Locally, this shear band only deforms the lower part of the SCSS, preserving the sedimentary unconformity at the base of the synorogenic unit (Alcañices syncline; González Clavijo, 2006). In other places the BLPD involves the entire SCSS with an upper and lower shear bands that limit lower-order shear band structures merging to create a first-order S/C structure (eastern Morais Complex; Dias da Silva, 2014). In many sections, deformation along the BLPD involves the BIMF deposits placed at the base of the LPa, thus resulting in a polygenetic mélange in the Festa et al. (2019) meaning. Also common is the presence of lower-order thrust duplexes within the LPa that are rooted in the BLPD. In these cases (S. Vitero Fm. in Alcañices Syncline, described in this section) the LPa is internally repeated by a stack of imbricated tectonic slices each with the SCSS at the base. The lensoid shape of the SCSS along the BLPD and associated structures also suggests that it was submitted to a strong tectonic pinching (thinning) and swelling (thickening) during thrusting. However, deformation is not always pervasive, as several size lenses of comparatively undeformed SCSS preserved fossilliferous content at many localities (Romariz, 1962, 1969; Quiroga de la Vega, 1981; González Clavijo et al., 1997; Piçarra et al., 2006b), defining all the Silurian graptolite biozones but lower Rhuddanian and upper Ludfordian stages as well as the Pridoli series (González Clavijo, 2006; Piçarra et al., 2006b).

The most extensive outcrop of the studied syn-orogenic rocks occurs in the periphery (structurally below) of the Bragança Complex (Figs. 2 and 4). To the E of this allochthonous complex, the core of the Late Variscan Alcañices Synform (Figs. 2, 4 and 5, cross sections 4 and 5 in Fig. 3) is formed by several LPa synorogenic units tectonically piled up in a number of imbricated thrust units ($C_2$) folded by a train of NW-SE-trending upright folds ($C_3$-$M_3$) (González Clavijo and Martínez Catalán, 2002; González Clavijo et al., 2012). From the top (more internal) structural position to the bottom (more external) the synorogenic units are named: Gimonde, Rábano, San Vitero, and Almendra formations. According to the maximum depositional age (MDA) obtained through detrital zircon U-Pb geochronology (Upper Devonian to uppermost Mississippian),

these formations include progressively younger rocks from the more internal to the more external ones, thus suggesting migration of the depocentre toward the relative autochthon coeval to the Parautochthon stacking (González Clavijo et al., 2012; Martínez Catalán et al., 2016).

The structurally highest Gimonde Fm. (Pereira et al., 1999, Meireles et al., 1999a; 1999b) is formed by finely bedded phyllites and metagreywackes, and scarce polymictic microconglomerate lenses containing exotic clasts and native intraclasts. Its age has been considered Upper Devonian on base of fossil plant debris (Teixeira and Pais, 1973) and palynomorphs (Pereira et al., 1999). However, detrital zircon ages (samples SO-7, SO-8, SO-9, SO-12; Martínez Catalán et al., 2016) imply an early Carboniferous (Tournaisian-Visean) MDA.

The structurally underlying Rábano Fm. (González Clavijo and Martínez Catalán, 2002), also called External Gimonde in Martínez-Catalán et al. (2016), comprises diverse lithologies being the most abundant a BIMF sequence displaying large exotic olistoliths of deformed rhyolites and dacites, felsic metatuffs, epiclastic rocks, white and grey quartzite, Silurian lydite (black radiolarite) and ampelite (carbonaceous shale), greywacke, phyllite, and limestone (Fig. 6A). The ages of these blocks (sometimes hundreds of metres in length) based on fossils and U-Pb zircon ages range from Furongian to Emsian (González Clavijo et al., 2016). At the uppermost Rábano Fm. a flyschoid sequence made of phyllite, quartzlitharenite, and local polygenic microconglomerate holds deformed exotic clasts and lithified intraclasts (Fig. 7A, B, and C); including plagioclase and volcanic quartz mineraloclasts and quartz shards indicating a nearby volcanic source (González Clavijo, 2006). Detrital zircon ages performed in this wild-flysch (sample SO-6) support a synorogenic nature and points to a Visean MDA (González Clavijo et al., 2016; Martínez Catalán et al., 2016).

The San Vitero Fm. (Martínez García, 1972) is a flysch made of up to metre thick phyllite and quartzlitharenite rhythms and local lenses of polygenic microconglomerates holding exotic clasts and lithified intraclasts (Figs. 7D, 7E, 8A and 8B) (González Clavijo and Martínez Catalán, 2002). This unit was considered Upper Devonian or younger based on fossil plant debris (Teixeira et al., 1973) but detrital zircon studies (samples SO-4, SO-5 and SO-13) support a Tournaisian MDA (Martínez Catalán et al., 2016).

The structurally lower imbricated thrust system hosts the Almendra Fm. (Vacas and Martínez Catalán, 1987) which is a calciturbidite made of phyllite and calcarenite rhythms up to several metres thick (González Clavijo, 2006). Local lenses of polygenic conglomerates and microconglomerates holding exotic clasts and pebbles, and lithified intraclasts were firstly described by Aldaya et al. (1976). Their lithologies include phyllite, sandstone, litharenite, quartzite, limestone, orthogneiss, rhyolite, and felsic volcanic tuff (Fig. 7F, G and H). Major blocks of lydite with Silurian graptolites, and limestones (Fig. 6B and C) containing abundant fossils (bioclasts of corals, sciphocrinoides, bivalves, gastropods, and tentaculites) have been identified (González Clavijo and Martínez Catalán, 2002; González Clavijo et al., 2016 and references therein). Conodonts found in calcarenite yielded a Lower Devonian age (Sarmiento et al., 1997). Detrital zircon studies in the Almendra Fm. indicated a Visean MDA (sample SO-14), thus supporting the Variscan synorogenic origin of this unit (Martínez Catalán et al., 2016).

The LPa emplaced at the eastern rim of the Morais Complex (Figs. 2, 4 and 5, cross section 6 in Fig. 3) was described by Dias da Silva (2014) as a turbiditic synorogenic sequence comprising two stratigraphic units (Travanca and Vila Chã Fms.). They consist of coherent primary units, broken beds units and lenses of block-in-matrix units displaying hectometre-size olistoliths. Clasts include native intraclasts and soft clasts, and exotic deformed lydite, ampelite, and quartzite. These units lack fossil-based ages. A palynomorph study performed by Dr. Gil Machado (in Dias da Silva, 2014) was unfruitful because of poor pollen preservation owing to Variscan thermal imprint. Detrital zircon studies (samples VC-21ZIR; VC-45ZIR; and VC-57ZIR) suggest a Devonian MDA (c. 390 Ma; Dias da Silva et al., 2015).

In the Marão Range, W of Vila Real (Fig. 2), the westernmost extent of the studied rocks, tectonic slices appertaining to the Pa comprise several turbiditic sequences (Mouquim, Canadelo and Santos Fms.) displaying rhythms of phyllite and greywacke with some intercalations of volcanic tuffs towards the top (Pereira, 1987). As they lie concordantly above the SCSS dated by graptolites (Piçarra et al., 2006b) it was proposed, without fossiliferous evidence, a Devonian age for the flyschoid units (Pereira et al., 2006). González Clavijo (2006) supported a correlation between these units and the San Vitero flysch, in the Alcañices synform, on base to the lithologies and the stratigraphic position, thus meaning a possible Tournaisian MDA. According our proposal of tectono-stratigraphic scheme, all the stacked pile must be considered as belonging to the LPa.

## 4 LPa synorogenic units: Youngest detrital zircon age populations of the LPa units

The known existence of synorogenic LPa units, partially encircling the GTMZ to the E, fostered the present research, aimed to recognize their possible extension to other areas of the GTMZ (Fig. 2). In this work 17 samples from areas surrounding the Bragança, Morais, and Cabo Ortegal complexes were used for zircon U-Pb geochronology (location of all samples in the map of figure SF1.1; coordinates in Table SF1).

The detailed description of the new zircon geochronology study is presented in the Supplementary File and in Supplementary Images 1 to 9. The complete dataset with the new U-Pb isotopic analyses is given in Supplementary Tables 1-17. The reference and complementary U-Pb zircon age datasets used in the Multidimensional Scaling (MDS) process and other statistical procedures are in Supplementary Tables 18 and 19.

In this section, the youngest zircon grain and zircon population ages of our new results are commented for the different parts of the synorogenic carpet and compared with published data.

The Meirinhos area, to south of the Morais Complex (Figs. 2, 4 and 5, cross section 4 in Fig. 3), was divided in two stratigraphic units: Meirinhos and Casal do Rato following Pereira et al. (2009) and Rodrigues et al. (2003). They were considered synorogenic flyschoid deposits bearing olistoliths of quartzite, phyllite, greywacke, felsic and mafic volcanic tuffs, limestone, ampelite and lydite by Pereira et al. (2009). However, different age, stratigraphic features and structure were proposed (Sá et al., 2014) based on the reappraisal of the Lower Ordovician trilobite *cruziana* ichnofossils in Armorican-type quartzites originally described by Ribeiro (1974) in this sector. Field data allow us to confirm the earlier proposal and the recognition of new synorogenic features as slump folds, disrupted beds, and olistostromes including hectometre-size lydite and quartzite

olistoliths (Fig. 6D and E). In both stratigraphic units, blocks, cobbles and pebbles of native (lithified intraclasts, soft pebbles) and exotic (quartzites, ampelite and lydite, felsic and mafic volcanic rocks, limestone) sources have been identified. Biostratigraphic ages of the olistoliths range from early Ordovician (trilobite tracks, Ribeiro, 1974; Sá et al., 2014) to Silurian (graptolites, Pereira et al., 2009). Two samples of medium-grained greenish greywacke belonging to the turbiditic sequence (CR-ZR-01, and MEI-ZR-01; see location, coordinates and geochronology of all geochronology samples in the Supplementary File) were collected in this area with a youngest single zircon (YZ) with Upper Ordovician age (439±8 Ma for CR-ZR-01 and 443±6 Ma for MEI-ZR-01) and MDA (i.e. concordant age given by the youngest zircon population in the 90 or 95% concordance interval; it may not include the YZ) of 467±4 Ma (CR-ZR-01) and 486±3 Ma (MEI-ZR-01) (Supplementary File; SI-1A and B). These ages do not support a synorogenic character nor a Lower Ordovician age for these siliciclastic rocks. However, the sedimentary features and the cartographic and structural continuity and correlation with the previously described LPa unit east of the Morais Complex (i.e. Travanca Fm. *in* Dias da Silva et al., 2015) make possible to propose a Mississippian MDA for these synorogenic siliciclastic rocks, with contribution of early Silurian, Middle Ordovician and Tremadocian (magmatic?) zircon sources.

West of Mirandela, around the village of Suções, there is a tectonic window (Rodrigues et al., 2010) partially controlled by late- or post-Variscan NNE-SSW subvertical faults (Figs. 2, 4 and 5, cross sections 3 and 6 in Fig. 3) which displays a turbiditic sequence attributed to the Devonian (Ribeiro, 1974) or to the Silurian with small patches of Lower Devonian siliciclastic rocks in the upper stratigraphic positions (Rodrigues et al., 2010). The MTMT was firstly mapped in this area by Rodrigues et al., (2010), tectonically separating the UPa overturned fold sequence from the LPa imbricated thrust complex. The proposed Silurian-Devonian age was based on graptolite assemblages preserved in the Silurian lydites (Piçarra et al, 2006b). Field and geochronological data allow us to infer that the LPa in this sector is also a synorogenic sequence, presenting the classical flyschoid features (Fig. 6F) (made of centimetre to metre beds of pelite and greywacke Rodrigues, 2008). Scarce Variscan detrital zircon grains (Famennian) were found in the greywacke layers of the flysch sequence, in the Upper Schists Fm. (MIR-41: YZ= 369±7 Ma, MDA= 497±5 Ma, SI-4A; AD-PO-49: YZ= 468±39 Ma, MDA=494±27 Ma, SI-4B; AD-PO-55: YZ= 444±26 Ma, MDA=488±16 Ma, SI-5A) and in the Culminating slates and greywackes Fm. (sample AD-PO-57: YZ= 372±6 Ma, MDA=488±16 Ma, SI-5B). The combination of field and geochronology data suggest that the graptolite-rich lydites belong to exotic olistoliths or to the SCSS at the base of the synorogenic tectonic slices. Detrital zircon ages and the lithostratigraphy of these stratigraphic units enable us to consider them as coherent primary units with olistoliths, overlying a tectonic mélange developed in the SCSS.

To the W, in the Vila Pouca de Aguiar area (Fig. 2), the LPa comprise several tectonically stacked flysch units limited by thrust planes (Ribeiro, 1974; Ribeiro et al., 1993; Noronha et al., 1998; Ribeiro, 1998; Rodrigues, 2008). All units are formed by turbidite sequences, with millimetre to metre thick interbedded pelites and greywackes (quartzwacke towards the base). The coarse-grained layers are very rich in plagioclase, thus suggesting a near-source (felsic) volcanic input into the basin (Rodrigues, 2008). Differently sized lens-shaped bodies of grey quartzite, black limestone, felsic metavolcanic rocks, and highly sheared SCSS are described within these units (Rodrigues, 2008). Our field research permitted us to define these LPa

formations as coherent primary units with block-in-matrix deposits exposing exotic fragments in a synorogenic convolute sediment tectonically repeated by imbricated thrust faults. These units are currently considered as Silurian-Devonian based on Silurian graptolites preserved in the lydites (Piçarra et al., 2006b) and by lithological comparation with similar formations to the east of the Bragança and Morais Complexes (Ribeiro, 1974; Noronha et al., 1998; Ribeiro, 1998; Pereira, 2000). The detrital zircon geochronology of one sample of lithoclastic coarse-grained arkose (AD-PO-48B) returned a Famennian–Tournaisian youngest single zircon and maximum depositional age (YZ: 355±34 Ma; MDA: 364±22 Ma; SI-3B).

At the northern edge of the Bragança Complex (Figs. 2, 4, 5 and 9, cross section 3 in Fig. 3), the Upper Allochthon high P/high T rocks tectonically overlay the LPa. Field work confirmed the absence of the UPa stratigraphic units and the discontinuous cartographic outline of the Middle and Lower Allochthon (Meireles et al., 1999a, 1999b). These units were tectonically thinned by an extensional shear zone or truncated by an out-of-sequence thrust system. In the upper structural part of the LPa, Ribeiro and Ribeiro (1974) noticed the presence different sized rock fragments in the stratigraphic sequence. They describe (i) epizonal fragments such as phyllite, quartz-phyllite, quartzite, felsic tuffs and rhyolites, ampelite and lydite; and (ii) meso-catazonal fragments of paragneiss (albite, chlorite and K feldspar), blastomylonite, and biotite-garnet gneiss. We have verified that these rocks appear as centimetre to hectometre-size olistoliths, being dispersed in a wider area than previously estimated (Fig. 6G, 6H and 10A). The olistoliths usually cluster within large mass-wasting deposits (BIMF) or occur as isolated bodies in the flysch sequence. The native rock blocks consist of fragments of consolidated fine- to coarse-grained greywacke beds among other siltstone/sandstone intraclasts and pelitic soft-pebbles. The exotic blocks include highly deformed lydites and ampelites, rhyolites, felsic tuffs, quartzites and limestones. Regional work in this area (Meireles, 2013) divides this sector LPa in four formations: Coroto, Rio de Onor, Soutelo and Gimonde, all of them bearing flyschoid characteristics and olisotliths (Figs. 4 and 5). In some of these units (Coroto and Soutelo) the limestone blocks yield upper Silurian to Lower Devonian crinoids (Meireles, 2013). Inherited Cambrian-Middle Ordovician acritarchs and lower-middle Silurian palynomorphs were also found in the Rio de Onor Fm. (Pereira et al., 1999). In Soutelo Fm., we also report the presence of a sandy/quartzitic olistolith with poorly preserved brachiopods of the "Lingulla" genus (Sofia Pereira and Jorge Colmenar *pers. comm.*), which are particularly common in the Lower Ordovician Armorican quartzite of the CIZ (Marão Range; Coke et al., 2001). Three samples collected in the Gimonde Fm. (Meireles et al., 1999a; 1999b) were used for geochronology of detrital zircon grains: GIM-ZR-01 (microconglomerate), EC-PO-293 and AD-PO-66 (Quartz-lithic sandstones). Although two of these samples have yielded Silurian (EC-PO-293: YZ= 426±44 Ma and MDA= 435±29 Ma; SI-2B) and Furongian (AD-PO-66: YZ=431±31 Ma and MDA=483±18 Ma; SI-3A) youngest zircon ages, one sample has Variscan detrital zircon ages (GIM-ZR-01: YZ=327±9 Ma and MDA= 354±3 Ma; SI-2A) confirming the results obtained in previous studies (samples SO-9 and SO-12 in Martínez Catalán et al., 2016), which allowed the characterization of the Gimonde Fm. as a Tournaisian-Visean synorogenic stratigraphic unit.

The Picón Beach at Cabo Ortegal Complex is placed east of the Ortigueira locality, in the Galicia northern coastline (Figs. 2, 4 and 5, cross section 1 in Fig. 3). There, the Loiba unit if the Río Baio Thrust Sheet (Marcos et al., 2002) is structurally onto the tectonically sheared (autochthonous) SCSS that defines the BLPD. This tectonic unit is formed by low metamorphic grade

flysch sequences and discontinuous lens-shaped BIMF. Its field aspects and structural relationships with the overlying and underlying tectono-stratigraphic units, led us to consider this unit as part of the Variscan synorogenic marine basin. The detrital geochronology of a fine-grained quartzite sample (PICON-2) has yielded a Tournaisian maximum depositional age (YZ: 350±7 Ma; MDA: 357±4 Ma; SI-6) thus supporting the extent of the LPa from northeast Portugal to the northern Spanish coast.

## 5 Magmatic zircon ages of the LPa olistoliths

In all lithostratigraphic units of the LPa exotic (extrabasinal) and native (intrabasinal) grains, clasts, pebbles and large olistoliths have been identified. Exotic rock fragments show pre-sedimentary mild to high deformation and metamorphic aspects, contrasting with the usually poorly deformed, low-metamorphic flyschoid sequence. The native fragments include soft pebbles and intraclasts, often showing syn-sedimentary deformation features (slump folds, boudinage, bedding disruption and convolute bedding of turbiditic aspect: Fig. 6D and E). The most consolidated sedimentary fragments suggest that they were recycled within the basin, as a wild-flysch (Ribeiro and Ribeiro, 1974; Aldaya et al., 1976; González Clavijo and Martínez Catalán, 2002; Martínez Catalán et., 2016). These fragments were considered a proof of its synorogenic character, and also evidence that the basin was fed from areas of the Variscan belt already deformed and metamorphosed (Antona and Martínez Catalán, 1990; González Clavijo and Martínez Catalán, 2002; Martínez Catalán et al, 2004; 2008). Complementarily, the fossil flora, fauna, and ichnofossil findings in the exotic olistoliths display ages from Lower Ordovician to Middle Devonian (Fig. 5). This wide range of biostratigraphic ages seems to indicate that fossil findings belong to rock blocks within a synorogenic turbiditic unit, which is confirmed by stratigraphic features and detrital zircon ages (González Clavijo et al., 2016). Trying to confirm this hypothesis, a U-Pb zircon geochronology study on volcanic rocks considered to represent olistoliths from the LPa was performed (Fig. SF1.1). We have complemented our study with the integration of already published U-Pb zircon age data from other volcanic olistoliths in the LPa (Farias et al., 2014; González Clavijo et al., 2016).

The complete description of the samples, their location and geochronology is presented in the Supplementary File.

### 5.1 New ages from magmatic olistoliths

As a complementary study of the detrital zircon research in the LPa flyschoid sequences, four volcanic rock olistoliths from the LPa in the Alcañices synform and in the northern edge of the Bragança Complex were sampled for U-Pb geochronology. Sample EC-PO-337 was picked northeast of Bragança in an olistolith made of low metamorphic grade foliated green metadacitic pyroclastic tuff from the Rábano Fm. This rock consists of volcanic quartz crystals and plagioclase fragments surrounded by a recrystallized tuffaceous matrix composed of fine-grained quartz, sericite and white micas defining the tectonic foliation in the olistolith. The youngest single zircon age is c. 435±40 Ma (Telychian), and the magmatic concordia age is 442±22 Ma (3 ages, 95% concordant; SI-8A) with important age populations defining inherited concordia ages at 471±14 Ma (8 ages), 484±16 Ma (6 ages) and 494±13 Ma (9 ages) (Floian to Furongian).

Sample EC-PO-419 was collected in the northern limb of the Verín-Alcañices Synform, to NW of the Bragança Complex in a low grade foliated medium-grained felsic metatuff representing an olistolith associated with others made of rhyolite, quartzite, lydite, quartzlitharenite and limestone (Fig. 6G) in a mass-wasting slide within the siliciclastic synorogenic sequence (Soutelo Fm. in Meireles, 2013). This sample presents a porphyritic texture composed of a highly foliated recrystallized aphanitic matrix made of sericite, fine-grained quartz, white micas, chlorite, biotite, surrounding volcanic quartz and plagioclase phenocrysts and shard fragments. The youngest single zircon age is 439±8 Ma and the magmatic concordia age is 442±12 Ma (5 ages, 95% concordant). Other age populations are Darriwilian (465±11 Ma, 6 ages) and Floian (474±13 Ma, 5 ages) (Supplementary File; SI-8B).

Sample PET-01 was grabbed at the Spanish-Portuguese border in the Rábano Fm., in an olistolith cluster extending from the northern edge of the Bragança Complex to the northern limb of the Alcañices Syncline. The sampled volcanic body is a grey rhyolitic tuff with disseminated sulphides, the lattest dyeing with reddish colour the rock when weathered. It shows a porphyritic texture made of a roughly foliated recrystallized matrix of sericite, fine grained quartz, white micas, chlorite retrogressed from biotite and irregular and cubic opaques. The phenocrystals are of sericitized plagioclase and volcanic quartz crystals showing embayments, broken crystals and shards. A concordia age of 494.1 ± 1.1 Ma was attained (lower Furongian) (SI-9A), supporting an olistolithic nature of this sample, as it was collected in an olistostrome of the synorogenic Rabano Fm. (González Clavijo, 2006).

Sample RAB-01, located at the north of the Alcañices village, was hand-picked in a metre size block of a weathered intensely foliated rhyolitic tuff representing an olistolith from the lower part of the synorogenic San Vitero Fm. Phenocrysts of plagioclase, sometimes fragmented, and volcanic quartz crystals and shards are sourrounded by a fine-grained recrystallized matrix of quartz, sercite, white mica and opaques. Structurally it belongs to a horse with the sheared SCSS at the base (Fig. 8A). This sample yields a magmatic concordia age of 476.0 ± 1.5 Ma (Floian) (SI-9B). Its position within an olistostrome stratigraphically higher than the graptolite-rich SCSS (Fig. 8A) excludes other plausible explanations as an interlayered pyroclastic flow, or a sill (González Clavijo, 2006; Gutiérrez Marco, Sá, and Piçarra *pers. com.*).

## 5.2 Published ages from magmatic olistoliths

In the Alcañices synform several olistoliths of felsic volcanic rocks have been identified; all of them of rhyolite to dacite composition, and often forming large clusters elongated NW-SE.

Previous research (González Clavijo et al., 2016) obtained an age of the Nuez olistolith (NUEZ; see location of all samples in the Supplementary File), one of the major blocks forming a several kilometers-long cluster included in an olistostrome inside the synorogenic Rábano Fm., towards the southern limb of the Alcañices synform. This block contains two volcanic facies: dacite lava and dacitic quartz-eyed tuff (Ancochea et al. 1988). The LA-ICP-MS U-Pb isotope analysis of magmatic zircons of a dacitic tuff sample returned a concordant magmatic age of 497 ± 2 Ma (lowermost Furongian).

In the northern limb of the same synform, the Figueruela dacite (COS-8), was dated by SHRIMP-II U-Pb analysis (Farias et al., 2014) yielding a magmatic concordia age of 488.7 ± 3.7 Ma (around the limit Furongian/Tremadocian). This igneous rock

was interpreted as a dacitic lava flow interlayered in the Paraño Group of the Schistose Domain or Parautochthon *sensu lato*. Our field work in the area has revealed that the Figueruela dacite belongs to a major cluster of olistoliths in a large mass wasting deposit, mainly composed of blocks of felsic lavas (dacite and rhyolite) and tuffs, also containing large quartzite and lydite lenses. In our reinterpretation the Figueruela dacite is an olistolith contained in a basal block-in-matrix unit placed below the San Vitero Fm. coherent primary unit and above the sheared SCSS. Thus, here we support that the Figueruela dacite belongs to the synorogenic LPa as previously stated by González Clavijo et al. (2016).

At the northern edge of the Bragança Complex, other significant volcanic body, the Soutelo rhyolite (COS-7), was dated by SHRIMP-II U-Pb analysis (Farias et al., 2014) yielding a 499.8 ± 3.7 Ma (upper Miaolingian) concordia age. The aforementioned authors have included the Soutelo rhyolitic lava in the so-called Paraño Group and considered it as volcanic event in the preorogenic sedimentary sequence of the (Upper) Parautochthon. Our field study disclosed the existence of an important cluster of olistoliths of diverse lithologies as lydite, grey quartzite, greywacke, limestone, rhyolitic lavas, and acidic pyroclastic tuffs, being the last two the most abundant types. Complementarily, this major block-in-matrix unit is placed on top of the mylonitized SCSS that defines the BLPD. For all these reasons we consider that the Soutelo rhyolite is also an olistolith inside the synorogenic LPa.

## 5.3 The possible sources of the magmatic olistoliths are in the UPa

The UPa unit, structurally below the Lower Allochthon as defined by Dias da Silva et al. (2014) in the eastern rim of the Morais Complex (Figs. 2 and 3), contains a late Cambrian to Silurian detrital sequence with minor limestones and voluminous volcanism (Pereira et al., 2000; 2006). The main volcanic events are, from bottom to top, the Mora felsic to mafic volcanic rocks (Mora Volcanics; Dias da Silva, 2014; Díez-Montes et al., 2015); the Saldanha gneiss (Ribeiro, 1974; Ribeiro and Ribeiro, 2004; Pereira et al., 2006, 2008) and a large felsic and mafic volcanic-sedimentary complex (Volcano Siliceous Complex of Ribeiro, 1974 and Pereira et al., 2006), laterly renamend to Peso Fm. (Dias da Silva et al., 2016; Díez-Montes et al., 2015). The Saldanha gneiss is a rhyolitic dome composed of fine to coarse grained porphyritic lavas and tuffs, intercalated in the Cambro-Ordovician terrigenous succession below the Armorican Quartzite (Algoso Fm.; Dias da Silva et al., 2014), while the Peso Fm. lays above it, in the highest stratigraphic positions of the UPa (Dias da Silva et al., 2016).

Previous zircon U-Pb geochronology of some representative bodies of those volcanic rocks (MOR-18ZIR; SAL-1ZIR; PR-1; PR-2, location on Supplementary file) at the eastern fringe of Morais Complex yielded Furongian (Mora Volcanics, 493,5±2 Ma), Tremadocian (Saldanha Volcanics, 484 2,5Ma) and Upper Ordovician (Peso Fm., 455-460 Ma) magmatic ages (Dias da Silva et al., 2014; 2016), evidencing episodic voluminous vulcanism along the stratigraphic record of the UPa. These ages are coherent with their stratigraphic position, thus supporting that the polyphasic pervasive deformation underwent by the UPa has not disrupted the original sedimentary architecture as the interlayered volcanic rocks keep the putatively primary chronological order (Díez-Montes et al., 2015; Dias da Silva et al., 2016). Although none of these U-Pb analysis included a study of the inherited zircon ages in these rocks, their magmatic ages were crucial to confirm the presence of rocks with ages similar to those found in the olistoliths, pointing to the UPa as the likely source of the Cambro-Ordovician volcanic olistoliths

in the LPa (González Clavijo et al., 2016), such as the Nuez, Soutelo (COS-7) and Figueruela (COS-8) hectometer-size magmatic olistoliths.

The two new samples from the NW Morais Complex UPa (P-381 and P-385, location in Supplementary File) were analyzed for magmatic and inherited zircon ages (Supplementary File; SI-7). Sample P-381 is an intensely folded rhyolite with quartz phenocrysts in a foliated sericite and white mica matrix. Its youngest single zircon age is 456±9 Ma and the youngest concordia

(magmatic) age is 461±4 Ma (Darriwilian; 6 ages, 95% interval; SI-7A). Although it was collected in a similar structural/stratigraphic position, sample P-385 (foliated dacite with quartz, feldspar and plagioclase phenocrysts in micaceous highly foliated matrix) seems to be older, with the youngest single zircon at 468±13 Ma and a concordia magmatic age of 475±5 Ma (Floian/Arenigian; 5 ages, 95% concordant; SI-7B). Both samples have inherited Tremadocian (c. 477-485 Ma) and Furongian (c. 485-497 Ma) concordia ages, which are the main zircon age populations in these rocks. Ages of both rocks

confirm the results obtained in previous studies, which attribute a Middle-Upper Ordovician age for the Peso Fm. of the UPa (Dias da Silva et al., 2016). These new results also show that the UPa is likely the major source of the 440-475 Ma magmatic olistoliths and of Cambro-Ordovician-Silurian detrital zircons in the LPa synorogenic basin, also supported by the age distribution plots and the MDS diagrams presented below (see also the Supplementary File).

**6 Discussion**

**6.1 Structural and stratigraphic meaning of the Lower Parautochthon synorogenic basins**

The studied samples contain detrital zircons coeval with the Variscan orogeny time span (400-320Ma). These results permit us to extend the LPa to areas with flyschoid sequences, some of them with broken beds, slump folds, BIMF, olistostromes and olistoliths, previously not recognized as Variscan synorogenic deposits. From rocks underlying eastern parts of the Bragança and Morais complexes, where the LPa was described by Dias da Silva et al. (2014; 2015; 2020), it may be continued to the

480 south following the GTMZ boundary through the Meirinhos (MEI-ZR-01 and CR-ZR-01) and western Mirandela (MIR-41, AD-PO-49, AD-PO-5 and AD-PO-57) zones to end in the Vila Pouca de Aguiar imbricated thrust system (AD-PO-48B) as shown in Figs. 2, 4 and 5. Farther south, west of Vila Real, in the Marão Range (Fig. 2) some structurally staked units contain sequences described by Pereira (1987, 1989) are likely to be correlated with the LPa, with a lower SCSS unit (Campanhó Fm.) overlain by turbiditic sequences (Santos, Canadelo and Mouquim Fms.) which have been lithologically correlated to some of

485 the LPa stratigraphic units in the Alcañices Syncline area (González Clavijo, 2006).

To the west of the Alcañices synform, north of the Bragança Complex, three samples of the flyschoid sequence (EC-PO-293, GIM-ZR-01 and AD-PO-66), two new volcanic olistoliths ages (EC-PO-337 and EC-PO-419) and data from another olistolith (COS-7, Farias et al., 2014), support the intrepretation of the LPa rocks in several stacked slices as the Variscan synorogenic sequence (Tournaisian age or younger). These slices contain glided blocks of upper Cambrian to Ordovician/Silurian volcanic

rocks (Fig. 10A), and also graptolite-rich Silurian lydites (Meireles et al. 1999a, 1999b; Piçarra et al., 2006a, 2006b) which preclude an *in situ* interpretation for the volcanic rocks. A similar arrangement was unveiled in the Alcañices Synform from

two new samples picked in olistoliths (PET-01 and RAB-01) plus two more from the literature (NUEZ, González Clavijo et al., 2016 and COS-8, Farias et al., 2014) yielding upper Cambrian to Ordovician magmatic ages. These rock blocks appear among other Lower-Ordovician-Lower Devonian fossil-bearing (meta)sedimentary olistoliths, which occur in a sedimentary unit of turbiditic nature containing Upper Devonian to Mississippian detrital zircon grains.

The Gimonde Fm. exposed in the Alcañices Synform and in the northern edge of the Bragança Complex (together some other local names: Soutelo, Rio de Onor and Coroto in Meireles, 2013), has cartographic continuity with the Nogueria Group at the Verín Synform and, according to our field observations, also with the lower tectonic unit of the Paraño Group underlying a thrust structure positioned at the base of the "Quartzitic" middle unit of the Paraño Group (Marquínez, 1984; Farias, 1990) as displayed in Fig. 2. Also, a coherent primary unit of decimetre-scale interbedded phyllites and greywackes was newly identified in several zones (Fig. 10B), including the presence of thick beds of greenish fine-grained lithic sandstone close to the town of Verín. Block-in-matrix phacoid bodies enclosing native and exotic blocks were also observed (Fig. 10C and D), some of them large enough to be considered olistoliths (mainly made of quartzite, lydite, black limestone, and felsic lava and tuffs). Towards the base, Silurian lydite and ampelite beds are frequent and strongly sheared, thus forming a tectonic mélange involving synorogenic sediments and the SCSS (Fig. 10E). As in the other nearby sectors, the Verín Synform area can be also envisaged as a complex imbricated thrust system, where the horses repeat the SCSS and the Nogueira Group. In the eastern part of the SW limb of the Verín Synform, in Portugal (Fig. 2 and cross section 3 in Fig. 3), the Nogueira Group correlated Lower Schist Formation (Pereira et al., 2000), which maintains the sedimentary aspects described above, but with coarser grained (meta)sandstones very rich in angular quartz and plagioclase grains, thus supporting a more proximal volcanic-rich source area. Also, the olistoliths identified in this area within the synorogenic sequence are made of UPa rocks, as they present polyphasic pervasive deformation characteristic of the tectonically overlaying unit (Fig. 10F, G and H), contrasting with the low deformation observed in the rest of the LPa sequence. According to the terminology adopted in this work (Festa et al., 2019, 2020) the basal detachment zone of this stratigraphic unit (possibly the BLPD) is made of imbricated tectonic slices mixing the BIMF deposits and the mylonitized SCSS, forming a polygenetic mélange.

The revision of the stratigraphy of the Nogueira Group and the lower unit of the Paraño Group in the Verín Synform allow us to propose a new interpretation for rest of the Paraño Group (Quartzitic and Upper units), which occupies the core of the synform. It may be correlated with the UPa according to the following points:

(i)    There is cartographic continuity with the UPa exposures that surround the Bragança Complex (Figs. 2, 3 and 4).

(ii)    A thrust fault underliying the Quartzitic Middle unit of the Paraño Group has been identified in this work in coincidence with places where Farias (1990) identified protomylonites and crenulation cleavage, which is here interpreted as the MTMT following Dias da Silva (2014).

(iii)    The Upper and Quarzitic units of the Paraño Group are made of low grade pervasively deformed detrital rocks like the sequence forming the UPa under the Morais and Bragança complexes (Nuño Ortea et al., 1981; Alonso, et al., 1981; Farias, 1990)n abd they contain volcanic interbedded bodies with similar geochemistry and U-Pb zircon ages (Nuño Ortea et al., 1981; Alonso, et al., 1981; Farias, 1990; Valverde Vaquero et al., 2007).

(iv) The Paraño Quartzitic unit (Farias, 1990), which delineate the hinge zone and limbs of the Verín Synform (Fig. 2), shows spatial continuity and can be lithologically correlated with the Algoso Fm. in Portugal, which is considered an Armorican type early Ordovician quartzite in the UPa (Dias da Silva, 2014; Dias da Silva et al., 2016).

(v) A volcanic body placed above the Lower Ordovician quartzite in the Verín Synform, the Navallo traquite, has a radiometric age of 439.6±5 Ma (uppermost Ordovician to Llandovery; Valverde Vaquero et al., 2007), younger but coherent with that of felsic metavolcanic rocks in the Peso Fm. of the UPa around the eastern and northern rim sections of the Morais Complex.

To the north, in the Cabo Ortegal Complex, the Rio Baio thrust sheet (Marcos et al., 2002) is structurally located under the allochthonous units and has been correlated to the Schistose Domain (Pa) below the Órdenes, Bragança and Morais complexes (Farias et al., 1987; Ribeiro et al., 1990; Martínez Catalán et al., 1997). The internal structure of Rio Baio thrust sheet is complex, deforming a greenschist facies detrital sequence which includes quartzites and volcanic rocks (Arce Duarte and Fernández Tomás, 1976; Arce Duarte et al., 1977; Fernández Pompa and Piera Rodríguez, 1975; Fernández Pompa et al.,

1976; Marcos and Farias, 1999). Among the latter, the main bodies are the Loiba dacites, Costa Xuncos rhyolites and Queiroga rhyolites (Arenas, 1984, 1988; Ancochea et al., 1988). The stratigraphic sequence of the Rio Baio thrust sheet was considered Silurian by the fossiliferous content of some beds (Matte, 1968; Romariz, 1969; Iglesias and Robardet, 1980; Piçarra et al., 2006b). Nevertheless, a field reappraisal considered those Silurian levels to be placed at the base of the Rio Baio thrust sheet (Valverde Vaquero et al., 2005), following the basal tectonic contact of the Schistose Domain in the Cabo Ortegal Complex.

The base of the Rio Baio thrust sheet was detached from the autochthonous CIZ by a fault developed preferably in the SCSS (here interpreted as the BLPD). Immediately above the BLPD, a low grade turbiditic sequence is exposed at the coastline, where the PICON-2 sample was collected (Figs. 2, 3, 4, 5 and Supplementary File) giving support for a Variscan synorogenic origin of this sequence and the ascription of the lower part of the Rio Baio thrust sheet to the LPa. Thus, under the Cabo Ortegal Complex a Parautochthon comparable with that of the Morais and Bragança areas exists, with a little deformed thin LPa unit

overlying a tectonic mélange developed in the SCSS, and the upper part of the Rio Baio thrust sheet above, representing the preorogenic UPa. An isotopic age obtained by Valverde Vaquero et al. (2005) in the Queiroga alkaline rhyolite (U-Pb TIMS, 475±2 Ma - Floian) supports this ascription by comparison with felsic volcanic rocks of the UPa around the Morais Complex (Dias da Silva, 2014; Dias da Silva et al., 2014; 2016; this work).

Based on the geochronology results of the 17 magmatic and detrital rock samples presented here and previously published

zircon age data we interpret that the LPa Variscan synorogenic sedimentary and structural unit forms a continuous tectonic carpet underlying the GTMZ separating it from the CIZ. The LPa is not observed in some reaches of the limit between the two zones because if existing, it has been hidden by late Variscan transcurrent faults or due to the intrusion of Variscan granitoids. Between the Cabo Ortegal and the Bragança complexes and in the northern Porto sector the available data from the literature does not conclusively support the existence of synorogenic sequences which could be endorsed to the LPa, and no detrital

zircon studies have been performed until now. Only the San Clodio Series may represent a link between the LPa of Cabo Ortegal and Trás-os-Montes, although it is younger and, although imbricated, it is not fully allochthon.

The strongly deformed SCSS present at the base of every LPa tectonic slice and frequently separated from the synorogenic sequence by a thrust fault rooted in the BLPD, must be considered a tectonic mélange in the meaning proposed by Festa et al. (2019; 2020) as it also incorporates tectonic blocks and olistoliths from the base of the synorogenic sequences. So, when the

related shear zone incorporates glided blocks (Figs. 8A, 8B and 9) it could be considered as a polygenetic mélange (Festa et al., 2019; 2020). We envisage the SCSS as a mixing unit sharing rocks of the LPa and the Autochthon, where the Silurian components were scraped off from the CIZ local uppermost sequence during the emplacement, a mechanism suggested by Ogata et al. (2019), Semeraglia et al. (2019) and Hajna et al. (2019) for mélanges formation. This offscraping mechanism is supported by the presence of Silurian rocks in the Autochthon (CIZ) at the eastern part of the Alcañices Synform (González

Clavijo, 2006) and at the Sil Syncline area (Martínez Catalán et al., 2016).

## 6.2 Provenance of the siliciclastic rocks and olistoliths in the Lower Parautochthon

We have used Multidimensional Scaling (MDS) (ISOPLOT-R by Vermeech, 2018) to compare the new and the already published data on the zircon age populations of the NW Iberia synorogenic basins, with potential sources within the Iberian

Massif terranes (see Supplementary File for more details, and complete U-Pb age datasets in Supplementary Tables 18 and 19). This approach has proven successful in the improvement of paleogeographic reconstruction models in the SW Iberian Variscan belt (Pereira et al., 2020a; 2020b) where the zircon age fingerprinting of possible sedimentary sources feeding the Devono-Carboniferous synorogenic marine basins indicate that sediments were derived from the continental margins of Laurussia and Gondwana, and from a "missing" Variscan volcanic arc (Pereira et al., 2012). Opposingly to the SW Iberia case,

the potential source areas of sediments and olistoliths of the NW Iberia synorogenic basin can be easily found in the nearby tectono-stratigraphic domains (Autochthon and Allochthon). Because synorogenic basins are usually fed from the neighboring surrounding (orogenically active) highs, there is no need to recur to other more distant Variscan sectors like the Laurussian domains of Meguma, Avalonia or Baltica, to accurately determine sedimentary provenance.

In this study, the zircon age data used for estimating possible source areas of NW Iberia synorogenic marine basins is a selection

of published detrital zircon U-Pb ages of pre-Upper Devonian siliciclastic rocks of the NW Iberian Autochthon and Allochthon. The data compiled by Puetz et al. (2018) and Stephan et al. (2018) were combined with other from more recent literature (Abati et al., 2007; Albert et al., 2015; Díez Fernández et al., 2010, 2012, 2013; Dinis et al., 2012; Fernández-Suárez et al., 2000, 2002, 2003, 2014; Gutiérrez-Alonso et al., 2003, 2015; Linnemann et al., 2008, 2018; Martínez Catalán et al., 2004; Naidoo et al., 2018; Pastor-Galán et al., 2013; Pereira et al., 2011, 2012a, 2012b; Shaw et al., 2014; Talavera et al., 2012, 2015; Teixeira

et al., 2011; Zimmermann et al., 2015) (reference samples in Supplementary Table 18). We have performed a quality test to the U-Pb isotopic data in each sample, recalculating all the zircon ages following the procedure used in our samples (see Supplementary File for complete description of the quality test and data selection). This dataset is used to fingerprint the zircon

ages in the source areas using MDS and compare them with the extensive data collection of the synorogenic siliciclastic rock samples, which was expanded in this work from 13 to a total of 24 samples. We have also included new zircon age data on volcanic rocks from the UPa (two samples: P-381 and P-385) and from large olistoliths in the LPa (four samples: EC-PO-337; EC-PO-419; PET-1; RAB-1) to compare their age spectra with the detrital zircon samples, thus tracing the source areas for some of the large olistoliths and the flysch sequence.

The age data of the possible source areas was selected according to a conceptual paleogeographic model for the Upper Devonian-early Carboniferous (as provided in Dias da Silva et al., 2015 and Martínez Catalán et al., 2016). Following the reasoning explained in Supplementary File, we define Source A samples as representative of the sources eroding from the peripheral bulge developed in the Autochthon (CIZ, WALZ-CZ and OMZ), but also including the UPa slice as one of the most distal section of North-central Gondwana passive margin. Source B reflects the NW Iberian Allochthon (GTMZ), defined as an accretionary complex built along the Devonian and emplaced onto the Autochthon during the early Carboniferous, forming the Parautochthon (UPa and LPa) as the lowermost tectonic sheet. The UPa is included in Source A because it was part of the Autochthon at the beginning of emplacement, although it was later incorporated to the Allochthon. So, Source A could have been at either side of the synorogenic basin margins, belonging to the peripheral bulge in early Variscan times (Late Devonian), and forming the GTMZ basal thrust sheet in the early Carboniferous, as the thrust front moved towards inland Gondwana. Source A samples were grouped by stratigraphic age, considering that the general stratigraphy of the Autochthon and UPa was not substantially imbricated by major Variscan thrust zones at time of its erosion. Oppositely, in Source B, the GTMZ Allochthonous Complexes were separated according to their tectono-metamorphic unit/domain (UA, MA or LA) with all samples ranging from not fully constrained Ediacaran to Lower Ordovician stratigraphic ages and belonging to the Galician allochthonous units (Malpica-Tui, Órdenes and Cabo Ortegal complexes).

Using the MDS diagram with both potential sources as the base of our provenance interpretation, we have plotted the new and published zircon age populations of the synorogenic siliciclastic rocks, adding the new magmatic and inherited ages of the Middle Ordovician-Silurian volcanic rocks collected as olistoliths in the LPa and in the upper stratigraphic units of the UPa.

The final MDS plot (Fig. 11, Supplementary File for detailed information) shows two main age clusters: Cluster 1 – "Upper Parautochthon Middle Ordovician-Silurian volcanism" characterized by synorogenic sediments with abundant Cambrian and Ordovician zircon grains, a high concentration of Middle to Upper Ordovician ages and minor amounts of Silurian and Devonian ages (Fig. 12); Cluster 2 – "Multiple Gondwana-derived sources" characterized by populations with a wide variety of sub-clusters (Groups 1 to 7, Figs. 11 and 13; Supplementary File), including the Autochthon and the UPa (Source A), and the Allochthonous Complexes (Source B). The defined groups show direct relation with the most probable sources (A and/or B), but they also represent different grades of sediment mixing and recycling (Fig. 11).

The MDS diagram (Fig. 11) and age distribution plots (Figs. 12 and 13) suggest that there is not a decipherable pattern in the provenance of sediments, both in time and space. It is possible to recognize provenance changes along and across the same stratigraphic units, sometimes sampled in different beds that are a few centimeters apart (samples MIR-41 and AD-PO-49). Our analysis evidence that zircon provenance varies, with sediments coming from both A and B sources at the same time

and/or arriving to different sectors of the synorogenic basin. Mixing patterns between age groups can also be noted, with sediment recycling leading to dilution of sources towards those more typical of Gondwana as, for instance, Group 7 and Cluster 1 showing a trend to Group 5. Reversely, younger samples plot closer to sources attributable to the Allochthonous Complexes, as shown by the trend of Group 7, from samples SO-14, SO-1 and SO-2 towards the Upper Allochthon reference population, apparently marking the progradation of the allochthonous wedge onto the NW Iberian Autochthon from the Tournaisian to the upper Visean.

These fluctuations suggest variations on the topographic highs surrounding the synorogenic basin at both margins (accretionary complex and peripheral bulge) in the Upper Devonian-early Carboniferous (Fig. 14). The tectonic activity that controls the basin shape and sedimentation was the cause to highly erosive, large-scale mass-wasting processes leading to the large olistolith-bearing BIMF deposits. The synorogenic marine sediments (cohesive flysch and BIMF) were gradually incorporated at the base of the accretionary wedge as a tectonic carpet, forming polygenic mélanges. The rapid frontal accretion of trench turbidites (Kusky et al., 2020) allowed their fast exhumation, leading to their recycling within the basin (wild-flysch).

## 6.3 Origin of Variscan detrital zircon grains

A contrasting aspect of NW Iberia synorogenic deposits with their equivalents in SW Iberia (e.g. Pereira et al 2014, 2020a, 2020b; Rodrigues et al., 2015; Pérez-Cáceres et al., 2017) is the scarcity of Variscan ages in the detrital zircon age populations in our study case. Only 14 detrital zircon samples out of 24 (11 new + 13 from previous works) have a minor population of Variscan zircons, comparing with the predominant Variscan zircon populations in most of the synorogenic formations in SW Iberia.

Notwithstanding, synorogenic Variscan zircon grains are represented in all studied formations of the LPa, independently of the MDS age cluster they belong to (the clusters mostly define the "old" zircon age population patterns; Figs. 12 and 13). Because there is no evidence of volcanic activity associated to the synorogenic marine deposits of the LPa, the source of the rare Variscan detrital zircons must be searched elsewhere. A possibility is the Upper Devonian-Carboniferous volcanism coeval with the development of the SW Iberia synorogenic basins (Oliveira et al., 2019a, 2019b; Pereira et al., 2020b) or in the Visean ash deposits laying within the CZ condensed marine synorogenic sediments (Merino-Tomé et al., 2017). But because zircon age populations of synorogrenic sediments include those identified in the volcanic and sedimentary olistoliths (Figs. 12 and 13), a more local derivation seems reasonable.

To identify the sources for the Variscan zircon grains we must check the main zircon forming events represented in the allochthonous complexes of the GTMZ, and in the underlying Autochthon (CIZ, WALZ, and less probably CZ and OMZ). The oldest Variscan zircon populations in the LPa range from c. 400 to 380 Ma, which can be related to the Lower-Middle Devonian high-P/high-T metamorphism in the Upper Allochthon (Gómez-Barreiro et al., 2006, 2007). An alternative contribution could be Emsian volcanism in the Autochthon, as that identified in the CIZ to the south of Trás-os-Montes, in the core of the Tamames Syncline (c. 395 Ma; Gutierrez-Alonso et al., 2008). A younger group, between c. 380-370 Ma may have its origin in metamorphic zircons growth during exhumation of the Upper Allochthon (Martínez Catalán et al., 2016, 2019)

and to a small amount, in prograde metamorphism of the Middle Allochthon (Pin et al., 2006; Arenas et al., 2007; Arenas and Sánchez-Martínez, 2015; Santos Zalduegui et al., 1996; and references therein). The age group in the range c. 370-360 Ma may derive from the high-P/high-T metamorphic rocks of the Lower Allochthon (Abati et al., 2010; Díez Fernández et al., 2011; Santos Zalduegui et al., 1995). These early Variscan ages are found only in the allochthonous complexes and never in the CIZ (Source A). Younger zircon sources with ages of c. 340-320 Ma are associated to low-P/ high-T regional tectono-

metamorphic event and to magmatic pulses at c. 340, 335 and 320 Ma (Martínez Catalán et al., 2003; Díez Férnandez and Pereira, 2016, Díez Fernández et al., 2017; López-Moro et al., 2017; Gutiérrez-Alonso et al., 2018; Dias da Silva et al., 2018). But zircons of this age interval have been found only in the Almendra Fm., the most external imbricate of the synorogenic deposits of the LPa, and in the San Clodio Series.

**7 Conclusions**

New results from field and geochronology studies on the Variscan (c. 400-320 Ma) hinterland synorogenic marine deposits of the Parautochthon in NW Iberia are presented. They surround large parts of the Galicia – Trás-os-Montes Zone (GTMZ), separating the Allochthon and Parautochthon from the structurally underlying Autochthon of the Central Iberian Zone (CIZ). The new data are useful for better understanding the relationship of the NW Iberia Variscan hinterland marine basin with the structurally underlying and overlying units. We show that the two units defined in the Parautochthon at the eastern rim of the

Morais and Bragança complexes, cover an area larger than previously estimated, being exposed from Cabo Ortegal (NW Spain) to Trás-os-Montes (NE Portugal). The existence of a preorogenic highly folded Upper Parautochthon (UPa) and an imbricated Lower Parautochthon (LPa) composed of slices of Devono-Carboniferous turbidites and tectonically scrapped autochthonous Silurian carbonaceous-siliceous slates (SCSS) is a general attribute of the whole NW Iberia Parautochthon. Regional tectonic and stratigraphic features show that the LPa represents a synorogenic basin that was gradually incorporated

into the base of the allochthonous accretionary wedge while it was being emplaced onto the northern Gondwana margin, forming a continuous tectonic carpet at the base of the GTMZ.

Analysis of the stratigraphic and tectonic features of the synorogenic flysch in the LPa highlights the relevance of large-scale mass wasting deposits in the sequence. Block-in-matrix formations (BIMF) represent sedimentary mélanges including large olistoliths derived from the accretionary wedge and probably the Autochthon exposed in a forebulge in front of the wedge.

They appear surrounded by a chaotic matrix with slump folds and broken beds of the flysch sequence. The BIMF formed due to gravitational instability triggered by tectonic activity within and at both margins of the basin. These deposits are frequently tectonized, forming imbricated thrust complexes with polygenic mélanges and tectonically scrapped autochthonous SCSS at the base.

Zircon geochronology of the LPa siliciclastic rocks and of magmatic olistoliths derived from the UPa and the Autochthon

constrain the provenance of the sediments and blocks in the LPa. Our study confirms the synorogenic nature of the LPa stratigraphic units, which include Emsian to Serpukhovian detrital zircon grains. These Variscan grains probably derive from

near sources located in the allochthonous complexes (metamorphic ages of 400-360 Ma), and from Variscan granitoids and migmatites (c. 340-320 Ma). A comparison of the detrital zircon populations of the NW Iberia synorogenic marine deposits, including the magmatic and inherited ages now obtained in the Middle Ordovician-Silurian volcanic rocks of the UPa (source) and LPa (olistoliths), with a compilation of reference samples from possible source-areas, allowed direct source-to-sink relationships of the Variscan hinterland basin with the accretionary complex (GTMZ) and the peripheral bulge in the Autochthon. Multidimensional Scaling analysis evidence intrabasinal sediment recycling and mixing of sources in time and space. These are explained by i) the tectonic instabilities within the basin and in its margins; ii) the migration of the depocenter towards inland Gondwana; and iii) the gradual incorporation of the synorogenic basin into the accretionary wedge leading to its denudation and recycling.

**Acknowledgements**

We appreciate the work of Kei Ogata (topical editor) and the valuable comments made by Francisco Pereira and John Wakabayashi (reviewers), which largely increased the quality of the text and figures. We also appreciate the constructive comments of Cristina Accotto that led to a deeper discussion of the data.

This work is a contribution to IGCP project 648, to project CGL2016-78560-P (MICINN, Spain), to project "Plan Cartográfico del IGME" (IGME, Spain), to IDL's (Portugal) Research Group 3 (Solid Earth dynamics, hazards, and resources) and to IDL's FCT-projects FCT/UID/GEO/50019/2019-IDL and FCT/UIDB/50019/2020-IDL. Í. Dias da Silva thanks the financial support by the SYNTHESYS3 (2015-2016) project DE-TAF-5798, by the FCT postdoctoral grant SFRH/BPD/99550/2014, and by the "Estímulo ao Emprego Científico – Norma Transitória" national science contract in the Faculdade de Ciências da Universidade de Lisboa (Portugal). JMC acknowledges a Program XII grant of the University of Salamanca that allowed a research stay at the Senckenberg Naturhistorische Sammlungen, Dresden, where some of the samples were dated. GGA acknowledges financial support from the Spanish Ministry of Science, Innovation and Universities under the project IBERCRUST (PGC2018–096534-B-100) and from mega-grant in accordance with Resolution of the Russian Federation Government (Agreement no. 14.Y26.31.0012).

**Data availability**

Data is available in Supplementary files, uploaded with this manuscript.

**Author contribution**

Emílio González Clavijo (EGC), Ícaro Dias da Silva (IDS), José R. Martínez Catalán (JMC), Juan Gómez Barreiro (JGB), Gabriel Gutíerrez Alonso (GGA) and Alejandro Díez Montes (ADM) have participated in the field work, and selection and preparation of the geochronology samples used in this work.

IDS, JMC, GGA, Mandy Hoffmann (MH), Andreas Gärtner (AG) and Ulf Linnemann (UL) have contributed in the of U-Pb-Th isotopic analysis and age data processing using LA-ICP-MS in the Senckenberg Geochronology lab in Dresden.

EGC, IDS, JMC, GGA were responsible by the elaboration of the manuscript, supplements, and illustrations, including field and microscope photos.

IDS was responsible by the geochronological analysis of the collected samples, the elaboration of the geochronological database and data quality test, and by the statistical analysis and sink-to-source correlation using Multidimentional Scaling.

**Competing interests**

There are no competing interests.

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

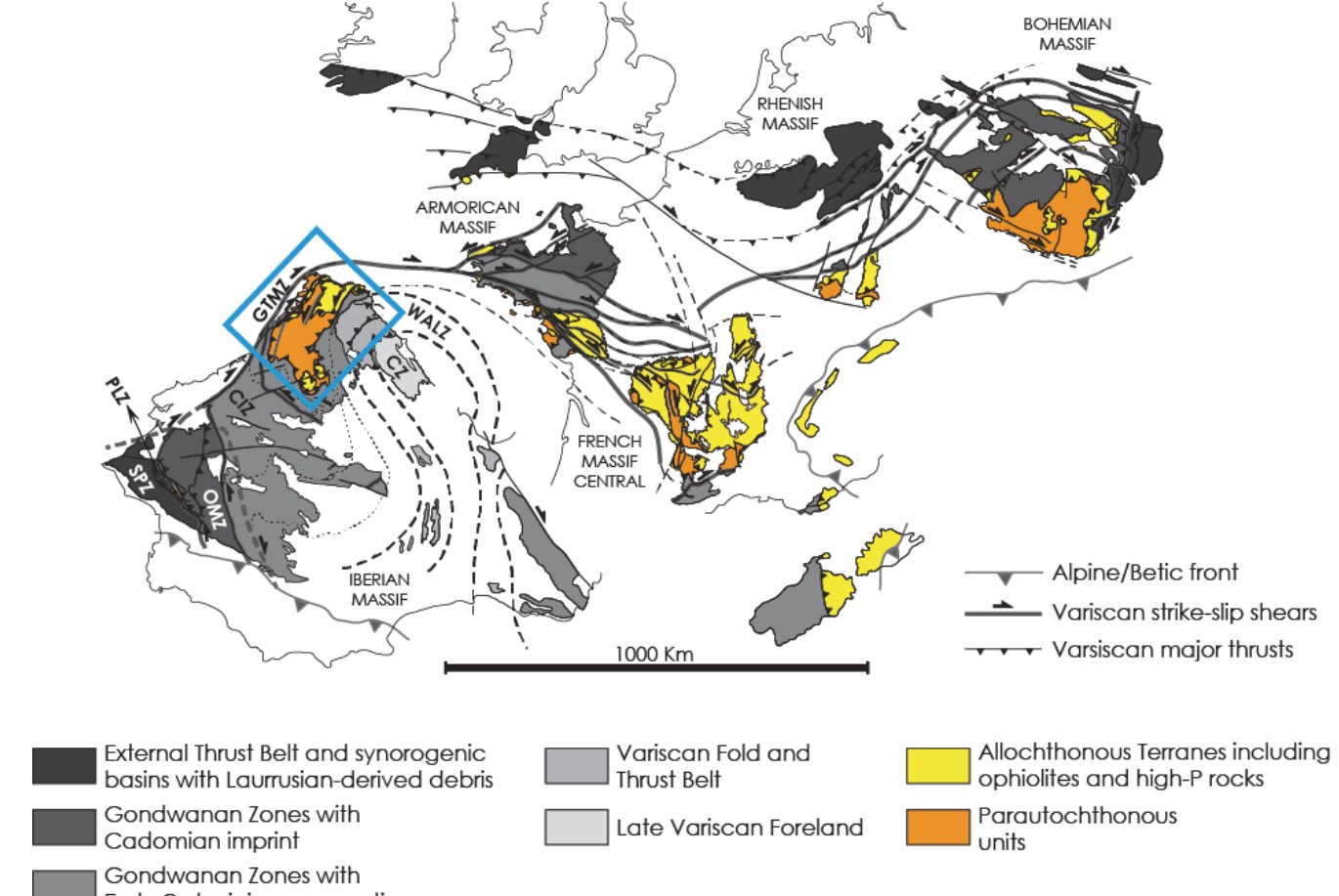

**Figure 1: Map of the European Variscan belt at the end of the Carboniferous. Modified from Martínez Catalán et al. (2007).**

 **Acronyms: CZ - Cantabrian Zone; WALZ - West Asturian-Leonese Zone; GTMZ - Galicia - Trás-os-Montes Zone; CIZ - Central Iberian Zone; OMZ - Ossa-Morena Zone; PLZ - Pulo do Lobo Zone; SPZ - South Portuguese Zone. Blue rectangle area is represented in Fig. 2.**

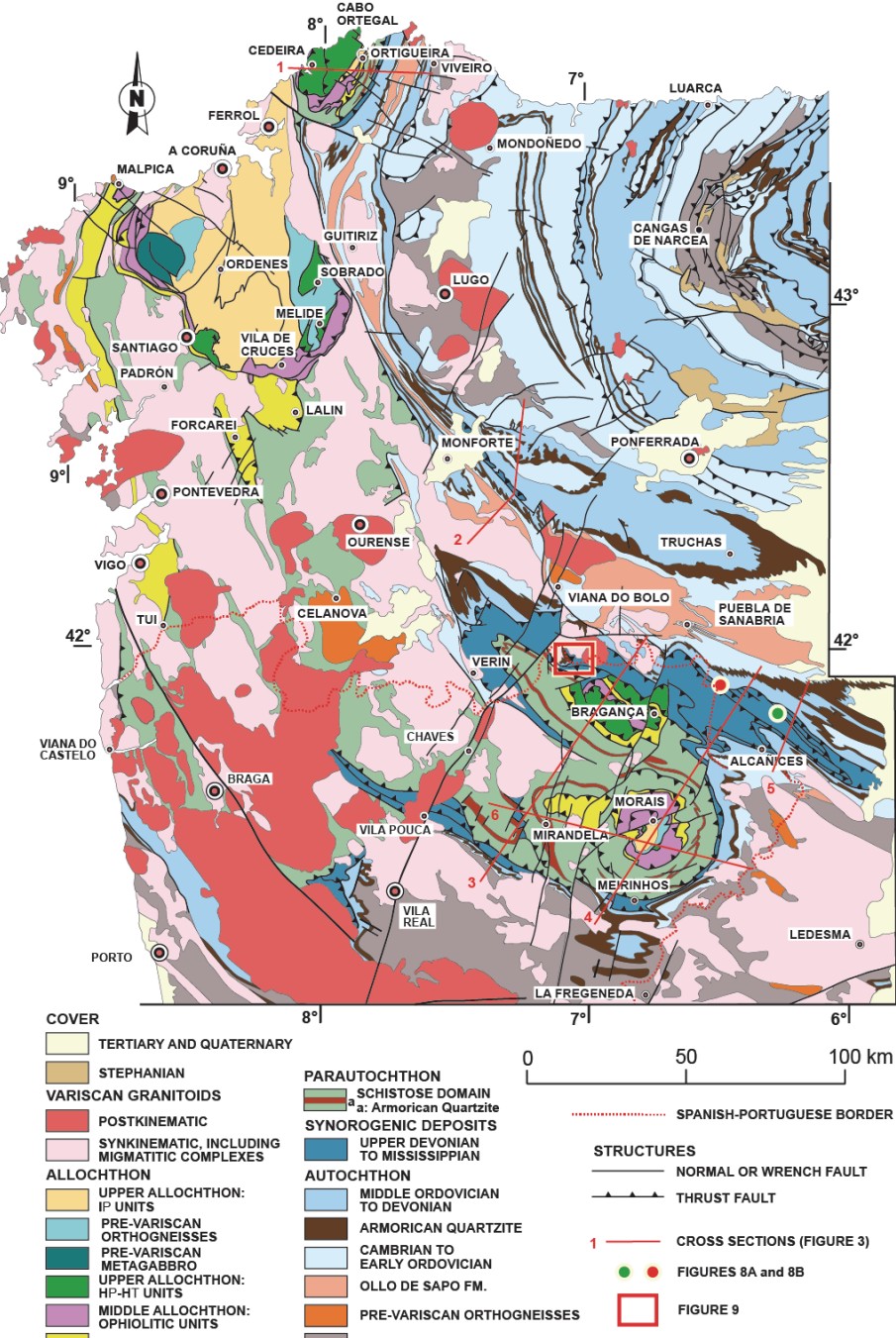

**COVER**

- TERTIARY AND QUATERNARY
- STEPHANIAN

**VARISCAN GRANITOIDS**

- POSTKINEMATIC
- SYNKINEMATIC, INCLUDING MIGMATITIC COMPLEXES

**ALLOCHTHON**

- UPPER ALLOCHTHON: IP UNITS
- PRE-VARISCAN ORTHOGNEISSES
- PRE-VARISCAN METAGABBRO
- UPPER ALLOCHTHON: HP-HT UNITS
- MIDDLE ALLOCHTHON: OPHIOLITIC UNITS
- LOWER ALLOCHTHON

**PARAUTOCHTHON**

- SCHISTOSE DOMAIN
  a: Armorican Quartzite

**SYNOROGENIC DEPOSITS**

- UPPER DEVONIAN TO MISSISSIPPIAN

**AUTOCHTHON**

- MIDDLE ORDOVICIAN TO DEVONIAN
- ARMORICAN QUARTZITE
- CAMBRIAN TO EARLY ORDOVICIAN
- OLLO DE SAPO FM.
- PRE-VARISCAN ORTHOGNEISSES
- LATE PROTEROZOIC (EDIACARAN)

SPANISH-PORTUGUESE BORDER

**STRUCTURES**

- NORMAL OR WRENCH FAULT
- THRUST FAULT
- 1 — CROSS SECTIONS (FIGURE 3)
- FIGURES 8A and 8B
- FIGURE 9

0     50     100 km

**Figure 2: Simplified geological map of the NW Iberia Variscan Massif modified from Martínez Catalán et al. (1997). The limits between Lower Allochthon, Upper and Lower Parautochthon and the autochthonous unit have been modified according to this work.**

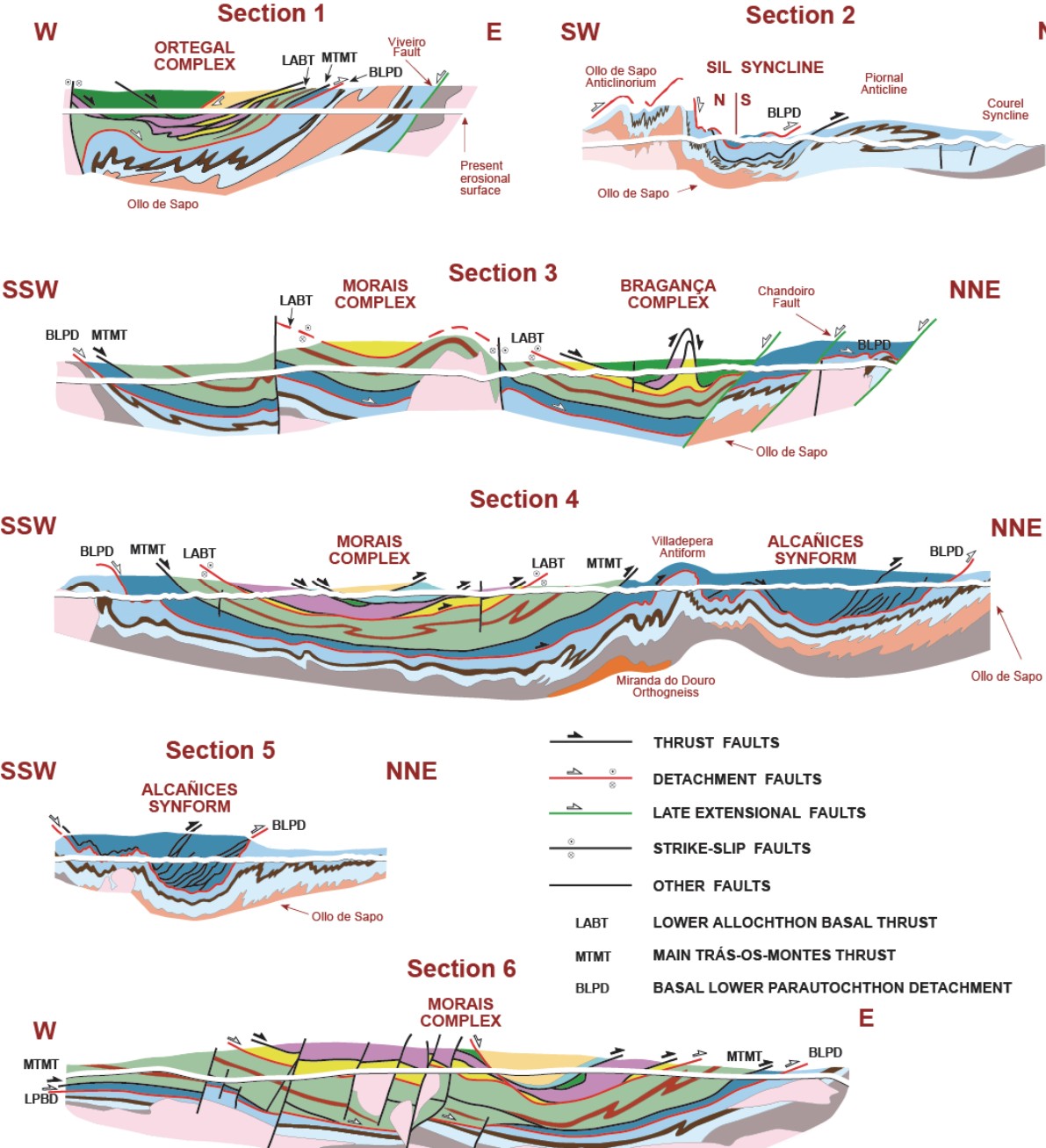

**Figure 3: Representative cross sections of the Variscan belt in the studied region including the inferred geometry of the synorogenic tectonic carpet as proposed in this study. See Fig. 2 for location and legend. Cross sections modified from: 1 – Marcos et al. (1984), Arenas (1988); 2 – Martínez Catalán et al. (2004); 3 – Pereira et al. (2006), Ribeiro (1974); Díez Montes (2006); 4 – Pereira et al. (2006), Ribeiro (1974), González Clavijo and Martínez Catalán (2002); 5 - González Clavijo and Martínez Catalán (2002); 6 - Pereira et al. (2006), Dias da Silva (2014).**

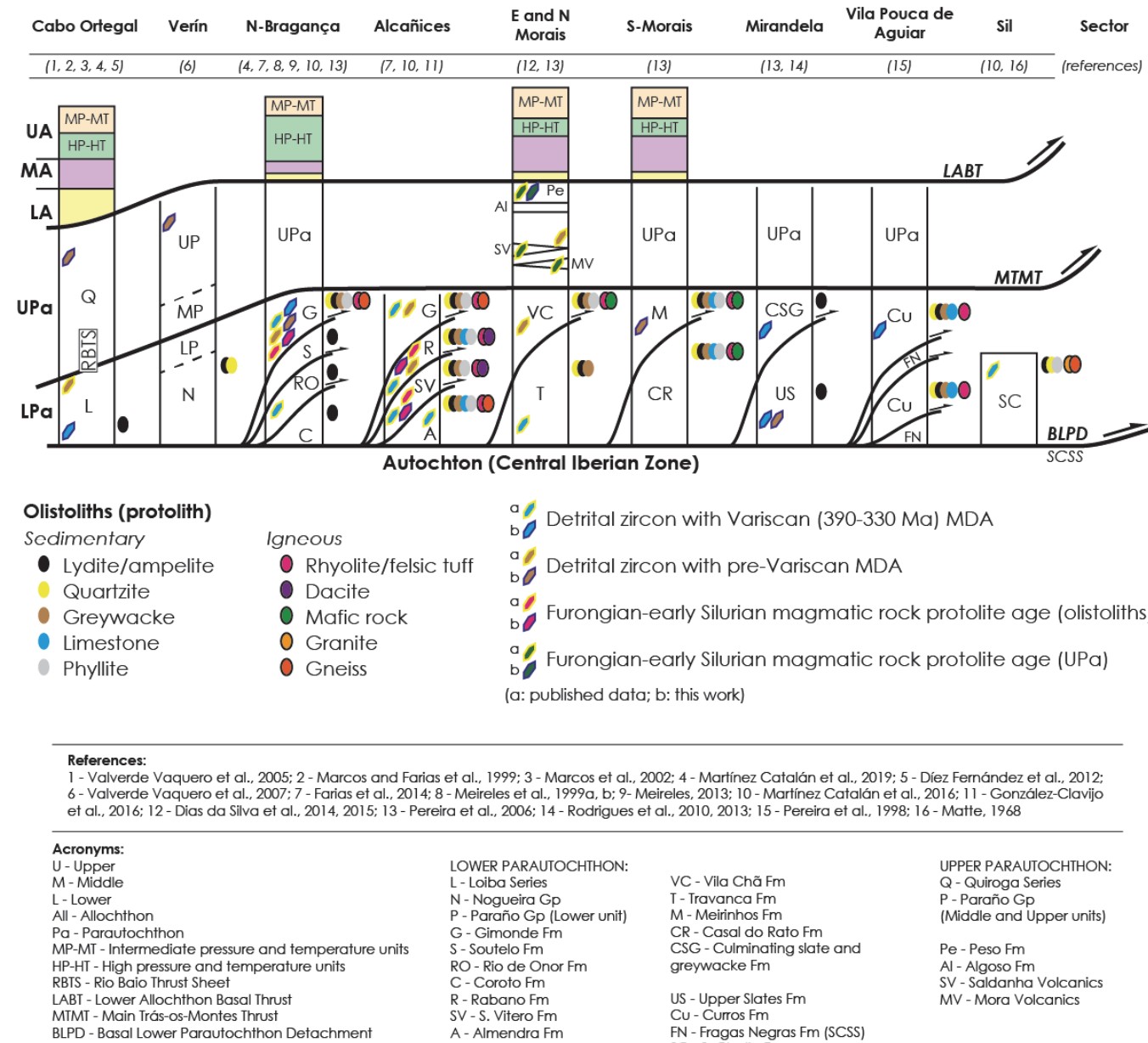

**Olistoliths (protolith)**

*Sedimentary*
- ● Lydite/ampelite
- ● Quartzite
- ● Greywacke
- ● Limestone
- ● Phyllite

*Igneous*
- ● Rhyolite/felsic tuff
- ● Dacite
- ● Mafic rock
- ● Granite
- ● Gneiss

a/b Detrital zircon with Variscan (390-330 Ma) MDA

a/b Detrital zircon with pre-Variscan MDA

a/b Furongian-early Silurian magmatic rock protolite age (olistoliths)

a/b Furongian-early Silurian magmatic rock protolite age (UPa)

(a: published data; b: this work)

**References:**
1 - Valverde Vaquero et al., 2005; 2 - Marcos and Farias et al., 1999; 3 - Marcos et al., 2002; 4 - Martínez Catalán et al., 2019; 5 - Díez Fernández et al., 2012; 6 - Valverde Vaquero et al., 2007; 7 - Farias et al., 2014; 8 - Meireles et al., 1999a, b; 9- Meireles, 2013; 10 - Martínez Catalán et al., 2016; 11 - González-Clavijo et al., 2016; 12 - Dias da Silva et al., 2014, 2015; 13 - Pereira et al., 2006; 14 - Rodrigues et al., 2010, 2013; 15 - Pereira et al., 1998; 16 - Matte, 1968

**Acronyms:**

| | | |
|---|---|---|
| U - Upper | LOWER PARAUTOCHTHON: | UPPER PARAUTOCHTHON: |
| M - Middle | L - Loiba Series | VC - Vila Chã Fm | Q - Quiroga Series |
| L - Lower | N - Nogueira Gp | T - Travanca Fm | P - Paraño Gp |
| All - Allochthon | P - Paraño Gp (Lower unit) | M - Meirinhos Fm | (Middle and Upper units) |
| Pa - Parautochthon | G - Gimonde Fm | CR - Casal do Rato Fm | |
| MP-MT - Intermediate pressure and temperature units | S - Soutelo Fm | CSG - Culminating slate and | Pe - Peso Fm |
| HP-HT - High pressure and temperature units | RO - Rio de Onor Fm | greywacke Fm | Al - Algoso Fm |
| RBTS - Rio Baio Thrust Sheet | C - Coroto Fm | | SV - Saldanha Volcanics |
| LABT - Lower Allochthon Basal Thrust | R - Rabano Fm | US - Upper Slates Fm | MV - Mora Volcanics |
| MTMT - Main Trás-os-Montes Thrust | SV - S. Vitero Fm | Cu - Curros Fm | |
| BLPD - Basal Lower Parautochthon Detachment | A - Almendra Fm | FN - Fragas Negras Fm (SCSS) | |
| SCSS - Silurian carbonaceous-siliceous slates | | SC - S. Clodio Fm | |

**Figure 4: Regional correlation of the NW Iberia LPa stratigraphic units, and their structural relationship with the tectonically overlying (UPa and allocthonous units) and underlying geologic domains (autochthon, CIZ). See Fig. 5 for more information.**

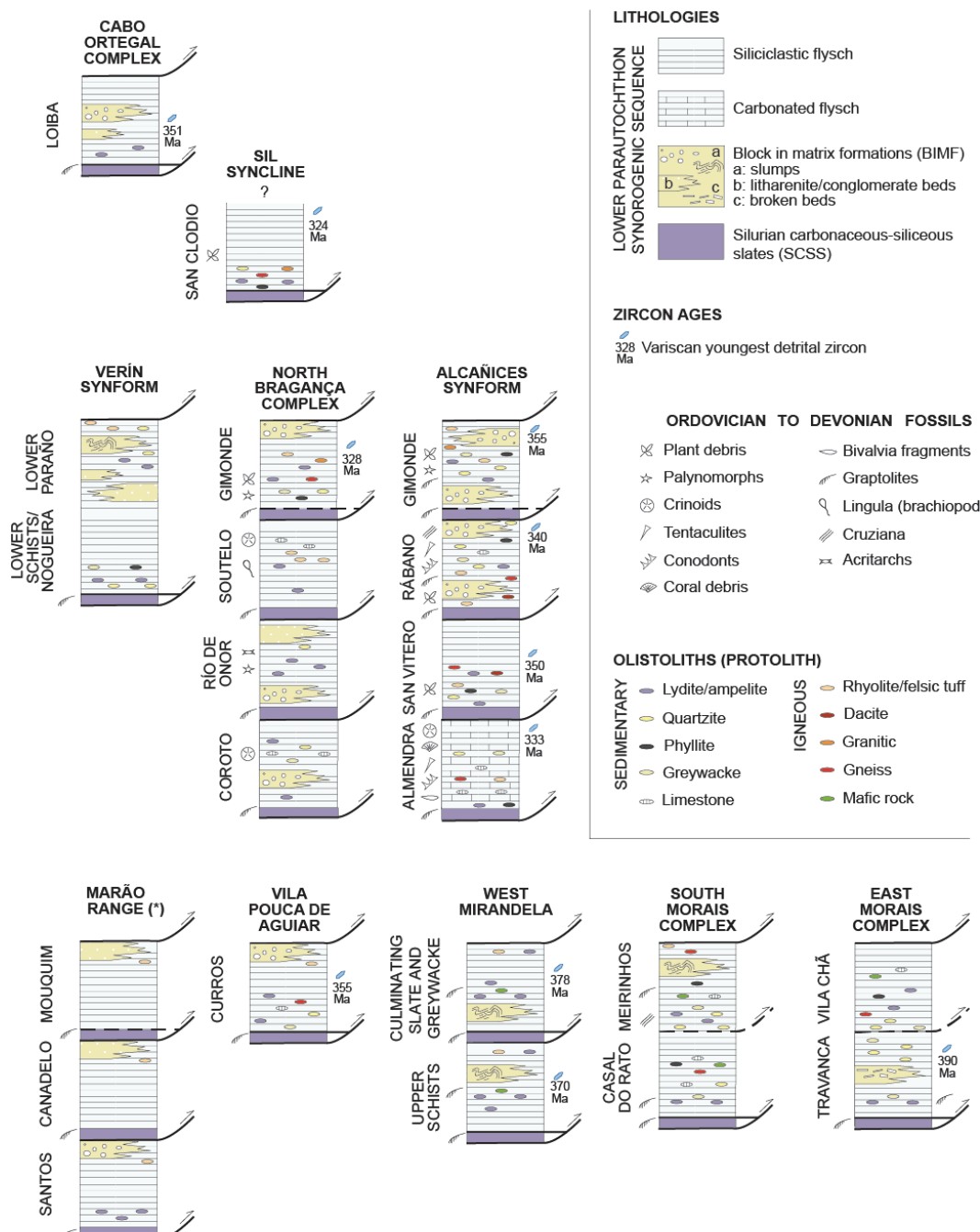

**Figure 5:** Synorogenic lithostratigraphic units of the Lower Parautochthon at the different sectors studied in this work. The sketch displays a simplified structure avoiding the minor tectonic slices repeating every stratigraphic unit. The sequences have been prepared considering all the data referred in the text and Fig. 4 for every sector and using the division and names that better fit with our field survey.

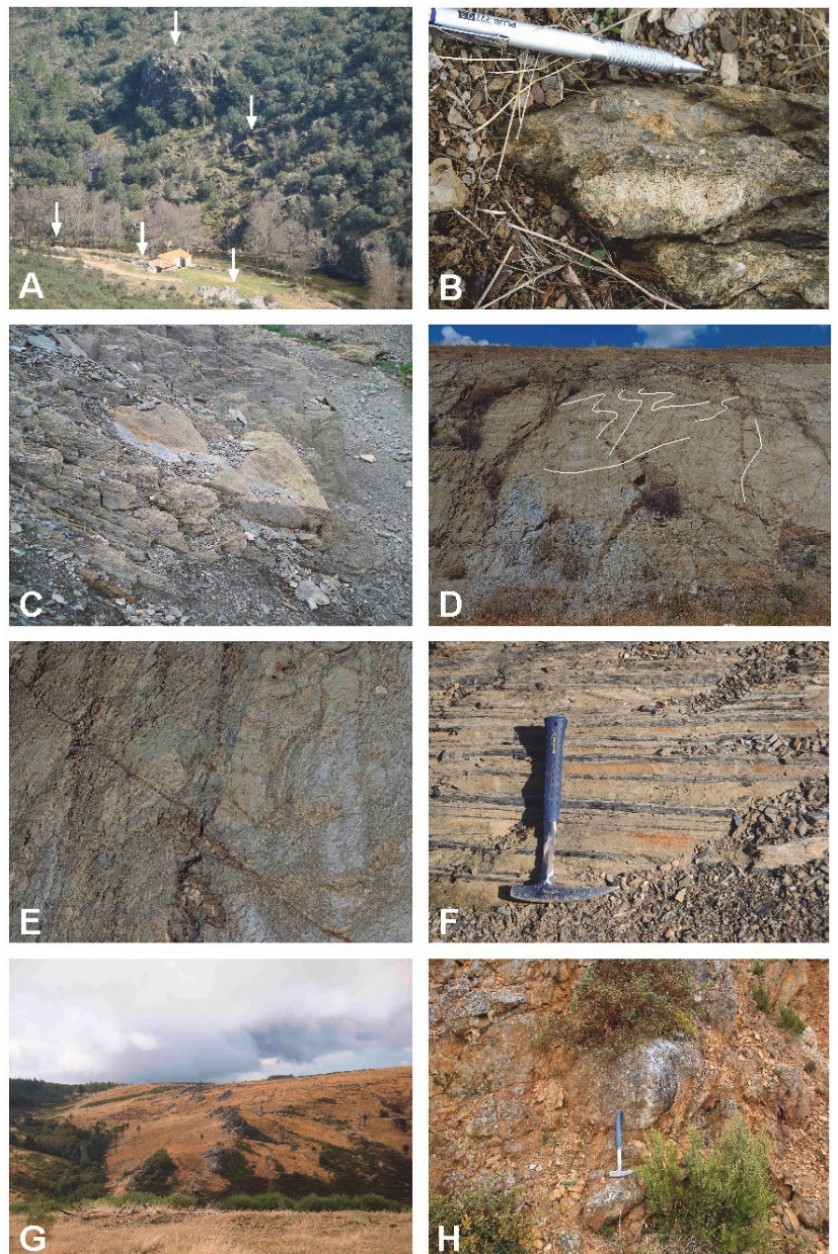

**Figure 6: Field aspects of the synorogenic sediments of the LPa: A) Lower Ordovician white quartzite olistoliths (arrows) in Rábano Fm; B) Centimetre size foliated rhyolite in an Almendra Fm micronglomerate; C) Metric glided block of limestone in an Almendra Fm grey phyllite bed; D) Slump folds in flyschoid facies at the Meirinhos synorogenic LPa; 2) Broken beds in a flyschoid facies of the Meirnhos Fm; F) Centimetre thick flysch facies in the LPa unit exposed in the tectonic window western Mirandela; G) Quartzite, lydite and rhyolitic tuff olistoliths in the LPa northern Bragança Complex; H) Blocks of quartzite (under the hammer) and lydite (above) in the LPa northern Bragança Complex**



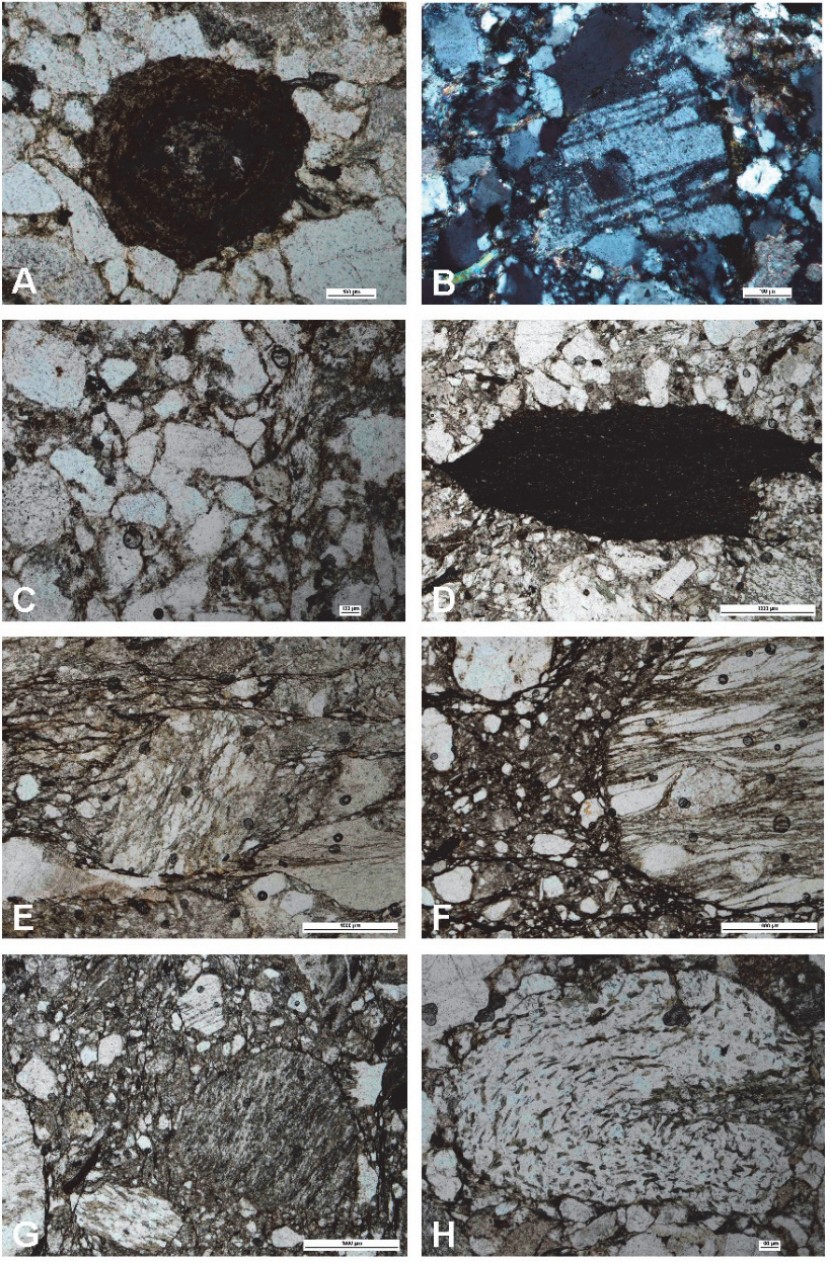

**Figure 7: Microphotographs of the sedimentary textures in the LPa formations, surrounding clasts of different natures. A) Bioclast in a Rábano Fm quartzlitharenite; B) Plagioclase mineraloclast in a Rábano Fm quartzlitharenite; C) Broken up volcanic quartz crystals in a Rábano Fm quartzlitharenite; D) Lithoclast made of black phyllite in a San Vitero Fm litharenite; E) Rounded clast displaying tectonic foliation almost normal to the sourronding external C$_1$ foliation (San Vitero Fm); F) Partial view of a rounded**

clast displaying mylonitic banding in an Almendra Fm microconglomerate; G) Randomly oriented clasts bearing previous foliation in an Almendra Fm microconglomerate; H) Rounded foliated clast displaying a microfold in an Almendra Fm microconglomerate.

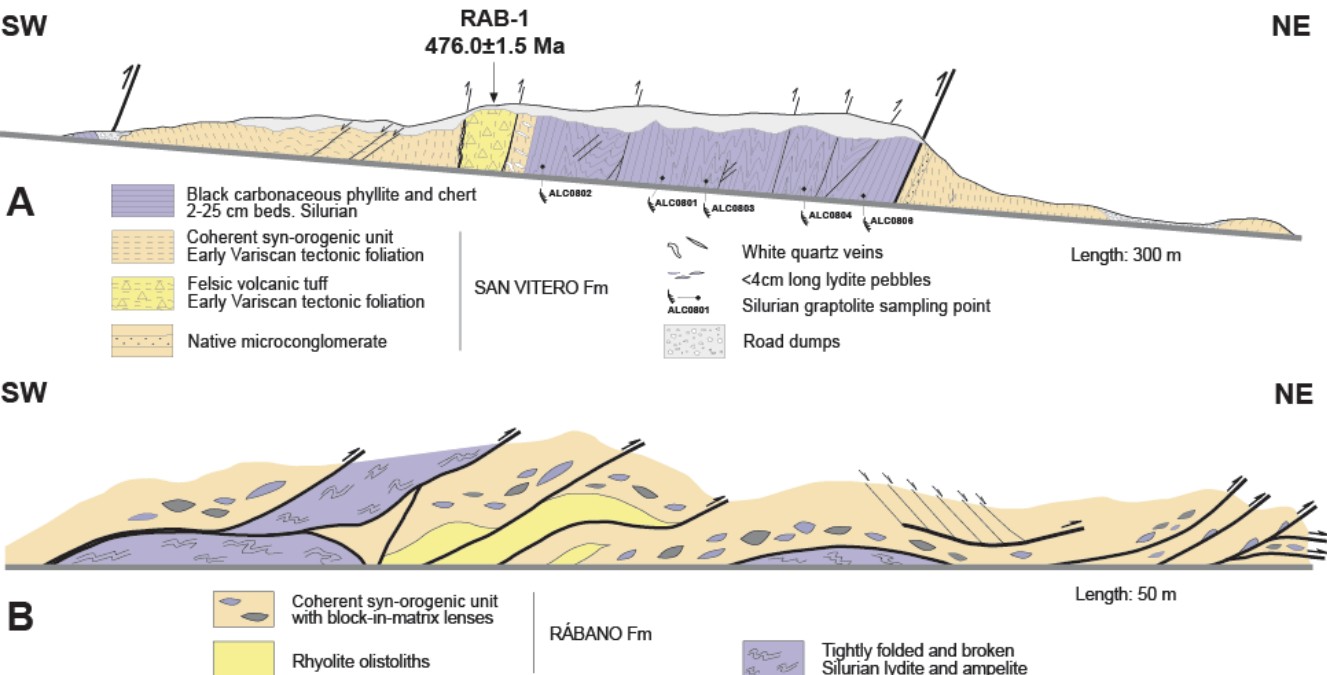

Figure 8: Examples of tectonic mélanges at the base of the synorogenic units in the Alcañices synform. A: San Vitero duplex slices
showing a Lower Ordovician volcanic olistolith above the Silurian fossiliferous sequence. B: Rábano Fm with rhyolite olistoliths of unknown age and displaying sedimentary block-in-matrix facies lately incorporated to a tectonic mélange.

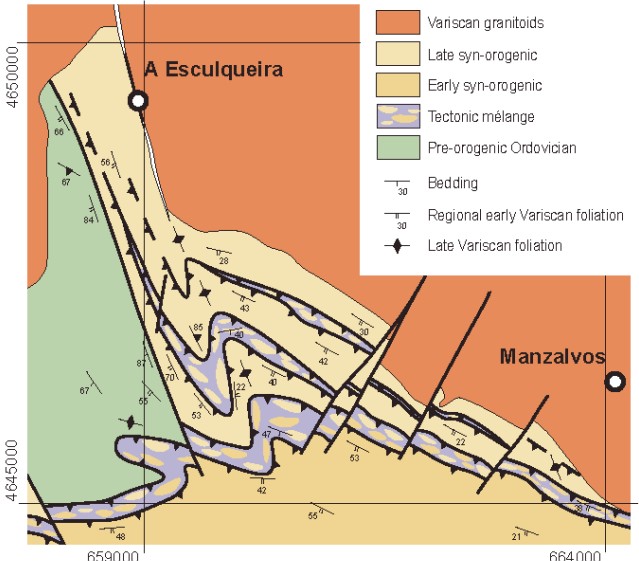


**Figure 9:** Geological sketch of one area N of the Bragança Complex showing slices of synorogenic units bounded by tectonics mélanges involving the synorogenic and the black Silurian rocks. The multiplex was openly folded during the Late Variscan event (C$_3$). Coordinates system: UTM-WGS84.

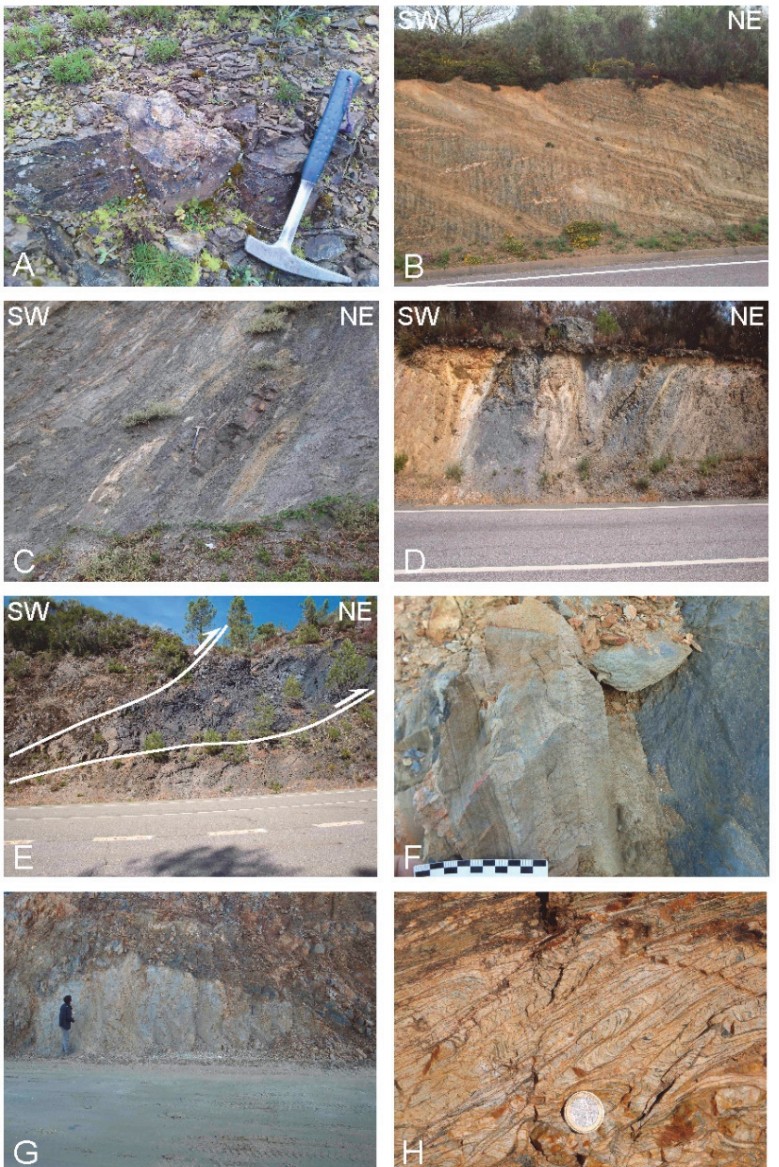

**Figure 10: Field aspects of the block in matrix formations and olistoliths in the LPa. A) Rhyolite block in the LPa northern Bragança Complex inside a quartzlitharenite; B) Flyschoid sequence in the Verín synform SW limb; C) Thrust band involving black Silurian ampelite, and rhyolite (whitish) and lydite (black) blocks in the NE limb of the Verín synform; D) Thrust band deforming rhyolite (yellowish and whitish), quartzlitharenite (brown in the right side) and ampelite (black) at the NE limb of the Verín synform; 3) Black Silurian condensed facies in a tectonic slice inside a tectonic mélange at the NE limb of the Verín synform; F) Flyschoid sequence with sedimentary and load structures in the LPa at the easternmost part of the Verín synform SW limb; G) Olistolith made of UPa rocks displaying the characteristic arrangement of tectonic foliations in the LPa of the easternmost part of the Verín synform SW limb; H) Tectonic foliation array of the UPa lower part at the SW of the Morais Complex.**

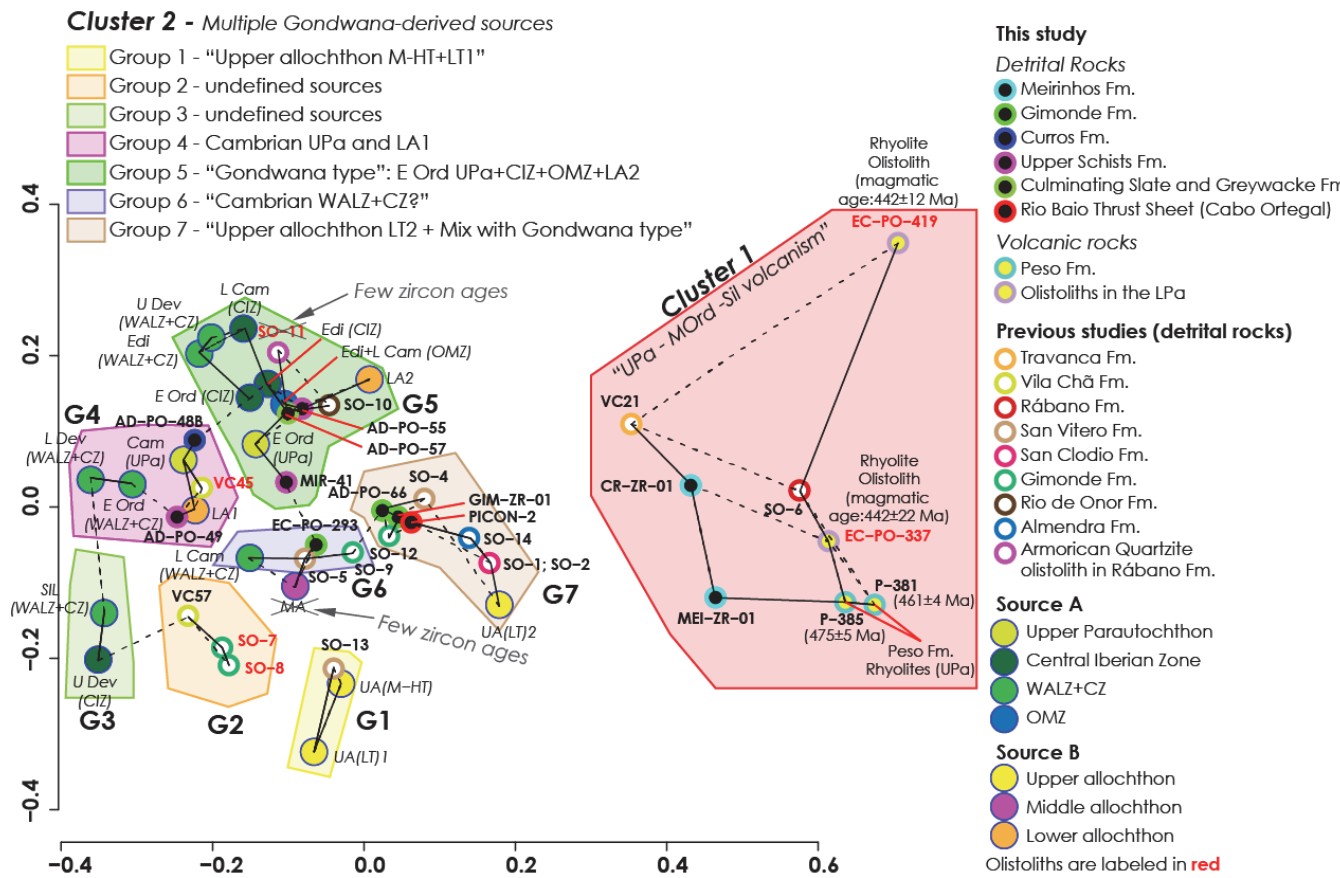


**Figure 11: MDS diagram for the studied samples and the reference populations in the autochthon and allochthon, with the differentiation between Cluster 1 (Fig. 12) and the seven groups of Cluster 2 (Fig. 13). Data to construct this plot is provided in the Supplementary tables 18 and 19. See more details on how this diagram was built in Supplementary File.**

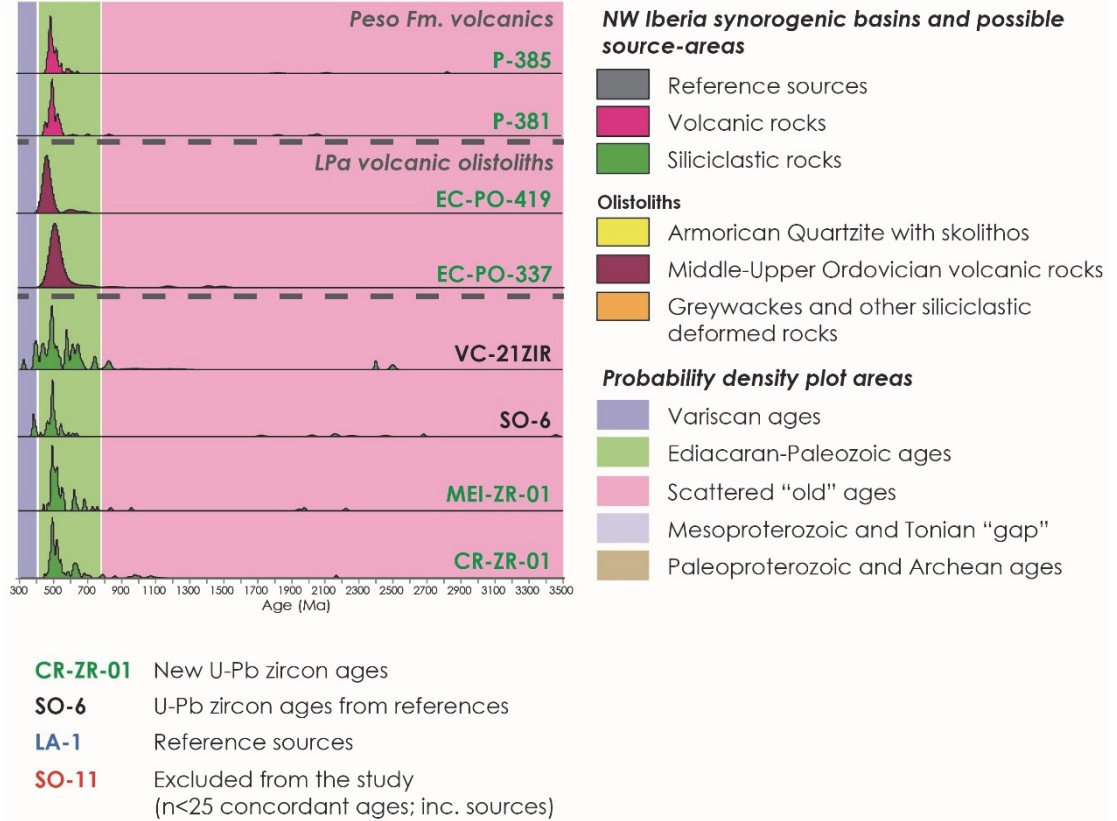


**Figure 12: Age distribution plots of the Cluster 1 type populations: "Upper Parautochthon Middle Ordovician-Silurian volcanism".**
**See text and Supplementary File for a detailed description.**

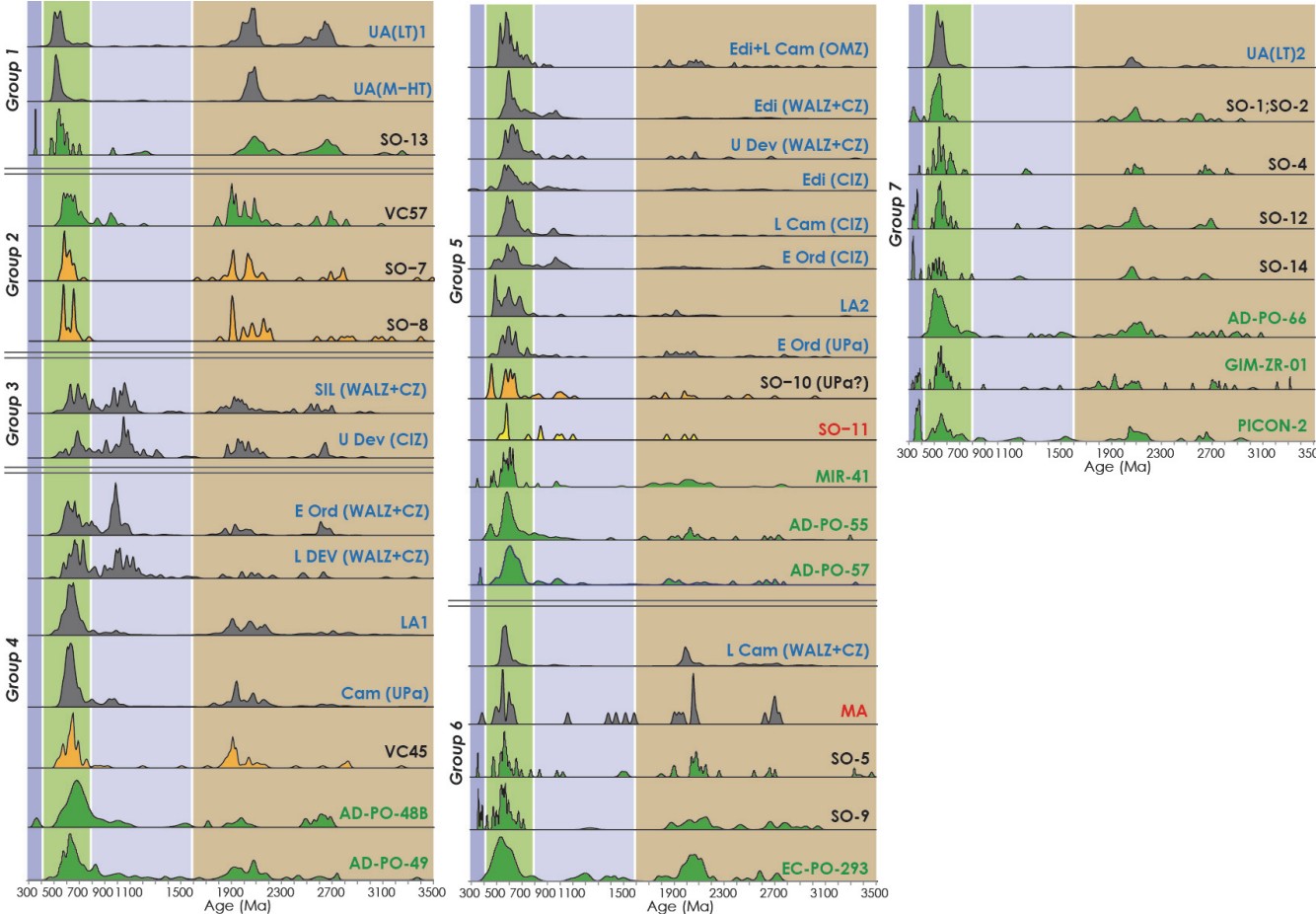

**Figure 13: Age distribution plots of all groups belonging to the Cluster 2 populations: "Multiple Gondwana-derived sources".** Legend is in Fig. 12. See text and Supplementary File for a detailed description.

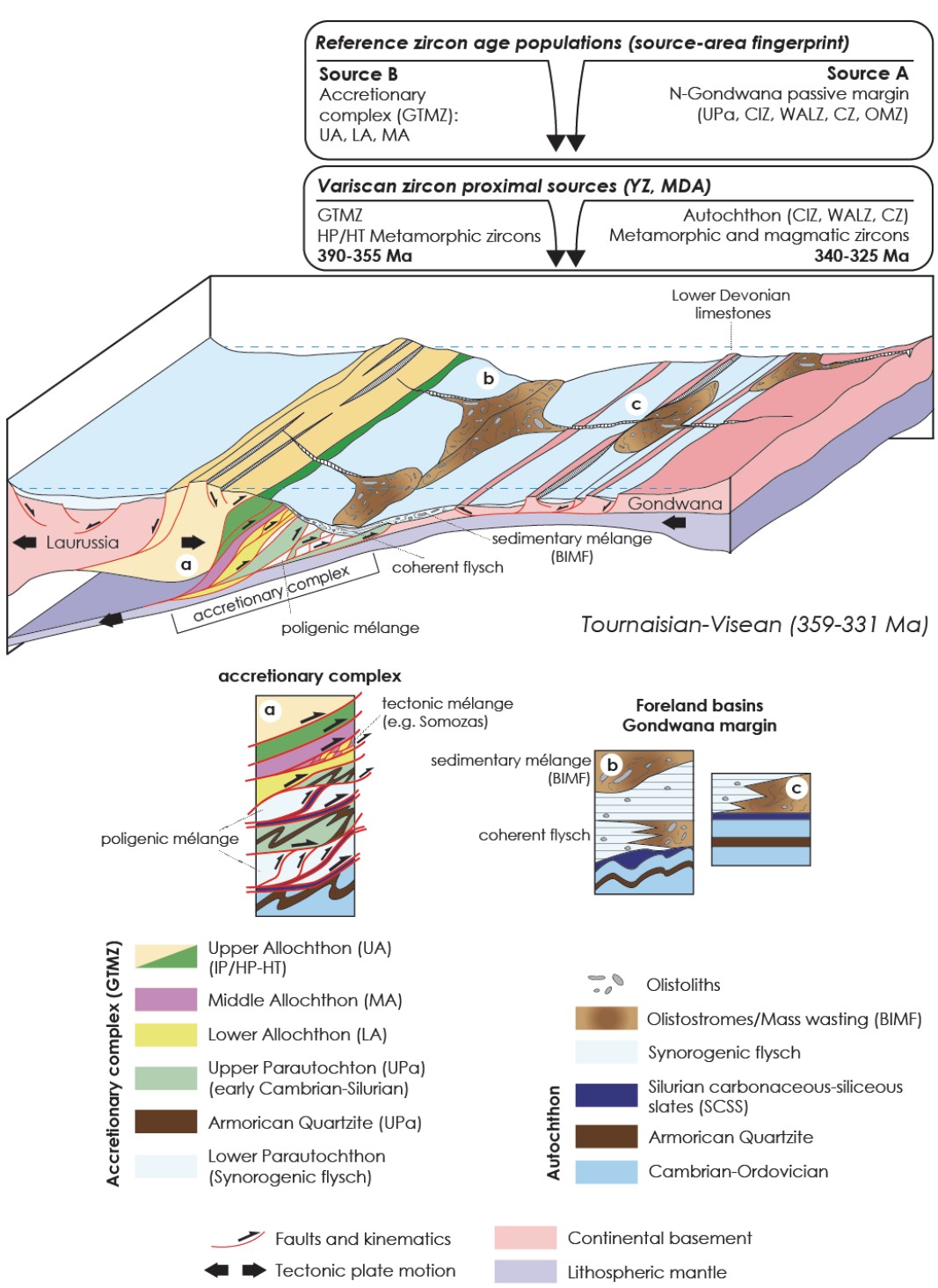

**Figure 14: Sketch of the orogenic collision at the Tournaisian-Visean displaying the trench-fill turbidites and the block-in-matrix deposits. The upper part represents the input of the zircon populations from different sources and zones.**