# Peer review of "A tectonic carpet of Variscan flysch at the base of a rootless accretionary prism in NW Iberia: U-Pb zircon age constrains from sediments and volcanic olistoliths"

_Solid Earth, 2020_

## Referee Comment (RC1) · Manuel Francisco Pereira (Referee) · 20 Oct 2020

The work of González Clavijo and co-authors on new field data and U-Pb zircon geochronology is relevant because it contributes to improving the interpretive models of the Variscan synorogenic sedimentation in NW Iberia. The new field data are convincing of the complex stratigraphy of these synorogenic deposits (Block-In-Matrix Formations- BIMFs) whose origin is associated with gravitational collapses instigated by tectonic instability during Gondwana-Laurussia accretion. The new geochronology data are of good quality and allow them to obtain magmatic ages of rocks that con-

stitute olistoliths and also to analyze the detrital zircon age populations of siliciclastic rocks (turbidites) that constitute the BIMFs. The comparison of the new geochronological data with a compilation of the existing data available in the literature allowed them to discuss the provenance of the BIFMs. I consider that the research line developed in this paper is well designed and the results are very interesting. However, I believe that the presentation and discussion of the data can be greatly improved considering this first version that was submitted. I consider that the organization of the text should be revised and English writing too (see my notes in attachment pdf file).

In any case, I would like to see more explored the following topic: The provenance of Silurian-Mid-Ordovician zircon grains found in the GTMZ Lower Parautochthonous units. They derived directly from magmatic rocks of Gondwana? or from Laurussia? If they derive from Laurussia, what Paleozoic terrain will they originate from? Meguma, West Avalonia, Ganderia, East Avalonia?

Please also note the supplement to this comment:
https://se.copernicus.org/preprints/se-2020-173/se-2020-173-RC1-supplement.pdf

**Supplement:**

[revised manuscript text omitted]

---

## Referee Comment (RC2) · John Wakabayashi (Referee) · 16 Nov 2020

Summary Comment: This interesting paper integrates U-Pb zircon data and field relationships to analyze detrital sediment sources in a synorogenic basin, as well as the progression of deformation that followed deposition. This appears to me to be a potentially valuable paper that will be of broad interest to researchers in orogenic processes and their geologic record. I suggest that a moderate level of revision is required for publication. In my opinion most of the suggested revision concerns improving the writing (English). In general, the English is of sufficiently high quality so that the scientific

meaning of the writing can be discerned through grammatical errors and non-standard word usage. However, the paper will be easier to read and have a greater impact if the English is improved. I have made some annotations on parts of the attached pdf, but these are by no means complete. These are examples that serve to illustrate that types of changes that should be made throughout the text, figure captions, and figure text labels. There are some technical comments in the annotations as well.

Additional (not among annotations in attached pdf) Technical Comment(s): Although describing block-in-matrix units is not the main goal of this paper, the field and petrographic observations are important because they are relevant to global debates on mélange origins. For this reason, I request that the authors add a bit more to their map scale, outcrop scale, and petrographic scale observations. Owing to the detail and quality of the existing observations, I suspect these additions can be made easily.

The types of observations I recommend are those that show the similarities and differences between tectonic mélanges and olistostromes. For example, the metamorphic assemblages in the rocks (if metamorphosed) should be described. Based on the existing descriptions it seems to me that some of the olistostromal blocks in the study area are higher in metamorphic grade than the matrix and that there is a range of metamorphic grade encompassed by the blocks in the olistostrome. Specific details should be given. In contrast it seems to me that the tectonic mélanges in the study area do not have blocks of higher metamorphic grade than the matrix or flanking units and there all of the blocks are isofacial, but the details are not given in the paper: they should be. These details are a subset of the larger relationship: exotic versus native blocks. The reason why I mentioned metamorphism, is that it can be difficult to tell if a block is exotic whereas, a block of higher metamorphic grade than flanking units or matrix is clearly so. In the paper it seems as if exotic blocks are confined to the olistostromes, whereas the tectonic mélange contains only blocks derived from the flanking units (or the specific disrupted zone) so that the blocks are entirely native. This should be clarified in the paper.

It seems to me that authors assume that readers will see the relationships summarized above as being clearly connected to origins as olistostromes or tectonic mélanges, hence they do not see the need to expand further on their observations. Yet there are many researchers who assert that the presence of exotic blocks is evidence for tectonic incorporation of blocks into matrix.

This brings up a still broader issue/concept, which is the definition of tectonic versus sedimentary versus "polygenetic" mélange. These terms are connected with the primary mode of mixing of blocks into matrix. In an olistostrome (sedimentary mélange), the blocks are mixed by sedimentary process, whereas a tectonic mélange, mixing of blocks into matrix (and creation of blocks) is a result of tectonic strain. For most readers, the meaning of "polygenetic" is not so easy to grasp in the sense of block-matrix relationships. It seems to me that many people mistakenly believe that this is simply the imposition of deformation on a sedimentary mélange but "polygenetic" means that additional blocks are created by deformation within and on the flanks of a sedimentary mélange: this can happen because of the creation of tectonic block creation by faulting of intact units bordering the sedimentary mélange and/or as a result of tectonic strain fragmenting some olistostromal blocks. A short explanation of the definitions of sedimentary, tectonic, and polygenetic mélanges should be given early in the paper, rather than simply referring to the definitions given in the cited papers.

Please also note the supplement to this comment:
https://se.copernicus.org/preprints/se-2020-173/se-2020-173-RC2-supplement.pdf

[Figure]

**Supplement:**

[revised manuscript text omitted]

---

## Author Comment (AC1) · 24 Dec 2020

Dear Manuel Francisco Pereira,

We gladly appreciate your comments and reviews to our manuscript. We are sure they will make it more interesting to read and more accessible for a broader audience.

After a close look to the annotated PDF you provided, we found that most of the proposed changes are related to English writing and other formal aspects. All of them were considered and are being applied to the revised version of the manuscript to be

uploaded. We went deeper in some comments and included some text parts which improve readability. Likewise, some text was eliminated to avoid repetitions and redundant text. We did not find necessary to make the text migration (e.g. section 5.2) as proposed by you, but not from John Wakabayashi (Reviewer 2).

We made some other important changes as you suggested. Fig. 4 and Table 1 migrated to the Supplementary File. We have produced 2 new figures: i) one (new Fig 4) with the tectono-stratigraphic correlation of the Lower Parautochthon (LPa) from Cabo Ortegal (Spain) to Marão Range (Vila Real, Portugal) showing the detrital and igneous zircon research made in the Parautochthon to date (including new data) and, ii) another (new Fig. 5) with the sedimentological aspects, main U-Pb (YZ and MDA) ages and biostratigraphic ages. They will be a great support for the text and undoubtedly will help the reader to follow the text. Because 2 new figures are added and tab. 1 and Fig. 4 migrated to the Supplementary File, we will also make changes in the figure numbering and their reference in the text. We also apply minor changes to some figures, according to the new figure order and to other graphical and grammatical aspects which we have identified and were not enunciated by any of the reviewers. Some moderate changes were made to Figure 13 (paleogeographic/source-to-sink model), namely we have erased the airborne zircon cloud from the (we discuss it in the manuscript) and corrected some aspects highlighted by the reviewers.

In relation to the discussion on the origin of the detrital zircons in the LPa, we believe that it is not necessary to have far away or occult sources, because all the pattern of the detrital zircon age populations in the synorogenic marine basin of NW Iberia - including the Variscan (400-320 Ma), Silurian, Upper, Middle and Lower Ordovician, Cambrian and older (Neoproterozoic-Archean) populations - are easily found in the neighbouring geological domains, in the underlying autochthon (representing the peripheral bulge) and in the tectonically overriding allochthonous complexes and Upper Parautochthon (representing the accretionary wedge), all considered "classical" Gondwanan sources. Because it is expected that in a synorogenic basin the source of sediments is generally

in the nearby (surrounding) reliefs, it is easier to assume that this particular segment of the early Carboniferous flysch basin cannot be compared with other basins such as the ones described in SW Iberia (South Portuguese Zone), where "exotic" laurrussian sources (i.e. Baltica, Meguma, Avalonia) have to be evoked to explain the presence of Early Devonian, Silurian, Upper Ordovician and Mesoproterozoic zircon age populations, which are absent in the nearby, currently exposed pre-Variscan basement (e.g. Ossa Morena Zone). We have dedicated some lines of text about this subject in the discussion section.

Once again, thank you for the comments and reviews!

Best wishes,

Ícaro Dias da Silva (corresponding author)

---

## Author Comment (AC2) · 24 Dec 2020

Dear John Wakabayashi,

On behalf of all authors of this paper I thanks your valuable revision. We gladly accept all the comments and suggestions you highlighted in both, the interactive comment and the annotated manuscript.

As in the case of Revisor 1, most of your comments were directed to the text organization and English writing. We have dedicated much time solving all the problems that

both of you have identified, and others that we recognized. For that, we have added or removed parts of the text, improving readability in many sections. Figures were updated and others were made new to follow the text. We expect that the manuscript will be easier to read and have a larger impact on the scientific community studying orogenic processes in the world.

The additional comments made by you in the interactive comment were very valuable to us. We have tried to incorporate most of your questions into the manuscript, but as you said, some of the raised subjects are a bit out of the scope of this paper. For instance, we would like to have a more detailed description of all the BIMF units we talk about, and their relationship in space and with the basement. However, we believe this subject deserves more attention and it should have its own space, in a dedicated work to the characterization of the large-scale mass-wasting deposits identified in the NW Iberia synorogenic basin. With this work we present the most complete geochronological study ever presented in Iberia to trace source-to-sink relationships based in the zircon population fingerprinting in the Variscan synorogenic marine basin and surrounding geological domains. Further works will certainly show other aspects not highlighted in this manuscript. Nevertheless, we have improved the text and some descriptions on the origin, metamorphic grade, deformation aspects in some of the studied olistoliths, highlighting the differences between flyschoid matrix, "exotic" extrabasinal and "native" intrabasinal rock blocks, resuming definitions of sedimentary, tectonic and polygenic mélanges in the beginning of the manuscript (as well as other definitions).

Once again, thank you for your dedicated review!

Best wishes,

Ícaro Dias da Silva (corresponding author)

––––––––––––––––––––––––––––––

---

## Author Response (AR1)

Ícaro Fróis Dias da Silva,
Montemor-o-Novo, 19th January 2021

Submission of the review to Solid Earth:

**A tectonic carpet of Variscan flysch at the base of a rootless accretionary prism in NW Iberia: U-Pb zircon age constrains from sediments and volcanic olistoliths**
By: Emilio González Clavijo, Ícaro Dias da Silva, José R. Martínez Catalán, Juan Gómez Barreiro, Gabriel Gutiérrez-Alonso, Alejandro Díez Montes, Mandy Hofmann, Andreas Gärtner, Ulf Linnemann

*Correspondence to*: Ícaro Dias da Silva (ipicaparopo@gmail.com)

Dear Kei Ogata, Topical Editor of Solid Earth,

First, we sincerely thank for all the comments and reviews made by Manuel Francisco Pereira and John Wakabayashi. Their valuable work helped to largely improve the quality of original version of the manuscript. We send you now the revised manuscript with most of the proposed changes accepted. However, there were some questions raised by the reviewers that we carefully answer in this letter.

We have attached 2 versions of the revised manuscript, one with Track Changes and another "clean" version with all the changes accepted. The document with Track Changes is below in this file. Some paragraphs were rewritten and improved following both reviewers' comments and other issues that we have found during the text and figure revision. You should easily follow the changes that we have made to the revised manuscript (with track changes) that we have uploaded.

Now, we will answer to all the comments from referees, point by point, discussing at the end the general comments they made in the review process.

**Comments from Referees and Authors response**

*R1 – Manuel Francisco Pereira*
**Introduction**

*Line 70: bad reference.*

Fixed

*Line 79: references missing.*

References added

*Lines 91-92: references missing.*

References added

*Line 91: Insert acronym.*

Changed to CZ

*Lines 95-105: Several minor text corrections.*

Text has been amended following the reviewer recommendations.

*Lines 109-110: Insert acronym.*

Changed to CZ GTMZ

**Geological Setting**

*Lines 112-125: Several minor text corrections.*

Text has been amended following the reviewer recommendations. Some parts have been rewritten to improve readability.

*Lines 126-135: Several minor text corrections.*

Text has been amended following the reviewer recommendations. Some parts have been rewritten to improve readability.

*Lines 139-140: I suggest the inclusion of a figure illustrating the GTMZ tectonostratigraphy.*

Two new figures (4 and 5) have been added to follow the text.

*Lines 141-165: Several text corrections.*

Text has been amended following the reviewer recommendations. Some parts have been rewritten to improve readability.

**Review of the synorogenic marine sequences in NW Iberia**

*Line 166 (section title): I suggest the inclusion of a figure illustrating the tectonostratigraphy and the relationship between distinct syn-sedimentary sequences (Sin syncline San Clodio Series, and the Alcanices syncline Gimonde, Rábano, San Vitero, and Almendra units), the underlying detachments and the occurrences of olistoliths, as well as, the LPa units (Travanca and Vila Chã Fms) and from the Marão Range.*

Two new figures (4 and 5) have been added to follow the text.

*Lines 167-196: Several text corrections.*

Text has been amended following the reviewer recommendations. Some parts have been rewritten to improve readability.

*Line 197: why do you use this terminology??? they are black slates and quartzites aren't they?*

We kept the original terminology because lydites correspond to black cherts and ampelites to graphitic slates. Text has been added to fix this problem.

*Lines 200-207: Several text corrections.*

Text has been amended following the reviewer recommendations. Some parts have been rewritten to improve readability.

*Line 208: lithified grains???*

We have clarified this issue. We added text to explain the difference between exotic (extrabasin) and native (intrabasin) clasts, including lithified (consolidated) pebbles and unconsolidated sediments (intraclasts).

*Lines 209-210: what do you mean???*

Text has been improved to clarify this sentence. Also, we have resumed all the text related to the shearing of the autochthonous Silurian unit (now called SCSS) at the sole of each Lower Allochthon tectonic slice.

*Lines 211-233: Several text corrections.*

Text has been amended following the reviewer recommendations. Some parts have been rewritten to improve readability.

*Lines 234-235: consider another description for this unit sheared unit of Black slates and quartzites.*

We have done it through the text. This unit is now called SCSS (Silurian carbonaceous-siliceous slates).

*Lines 236-238: Some text corrections.*

Text has been amended following the reviewer recommendations. Some parts have been rewritten to improve readability

*Lines 239-241: It would be important to illustrate this statement with a figure showing possible correlations between distinct units...*

Done, in the new figures 4 and 5.

**Extending the Lower Parautochthon**

*Line 242 (section title): LPa synorogenic units: U-Pb zircon geochronology.*

We have changed the section title to "4 LPa synorogenic units: Youngest detrital zircon age populations of the LPa units"

*Lines 243-250: Minor text corrections.*

Text has been amended following the reviewer recommendations. Some parts have been rewritten to improve readability.

**Stratigraphic sequences and youngest zircon ages**

*Line 251 (sub-section title): Youngest detrital zircon age populations of the LPa units.*

We have concluded that this section header was not necessary, so we had deleted it.

*Lines 252-269: Several text corrections.*

Text has been amended following the reviewer recommendations. Some parts have been rewritten to improve readability.

*Lines 270-315: I suggest that the text should be rewritten until the end of this section.*

Text has been amended following the reviewer recommendations. Some parts have been rewritten other deleted, to improve readability.

**Lower Parautochthon magmatic olistoliths, their ages and possible source-areas**

*Line 316 (section title): Magmatic zircon ages of the LPa olistoliths.*

We have changed the title according to the reviewer comment.

*Lines 317-328: Several text corrections.*

Text has been amended following the reviewer recommendations. Some parts have been rewritten to improve readability.

**Magmatic olistoliths age results**

*Line 330 (sub-section title): Delete.*

We have changed the title to: "New ages from magmatic olistoliths"

*Lines 331-354: Several text corrections.*

Text has been amended following the reviewer recommendations. Some parts have been rewritten to improve readability. We kept the description of the rhyolites, as proposed by the second reviewer (John Wakabayashi).

**Olistoliths magmatic ages from references**

*Line 356 (sub-section title): Delete.*

We have changed the title to: Published ages from magmatic olistoliths

*Lines 357-380: Several text corrections.*

Text has been amended following the reviewer recommendations. Some parts have been rewritten to improve readability.

**The possible sources of the magmatic olistoliths are in the UPa**

*Line 356 (sub-section title): Delete.*

We have kept the title heading.

*Line 357: this section (line 383 to line 407) can be moved to line 513 of section 6.2.*

After long thinking and discussion we decided to keep this sub-section in its position in the manusctipt.

*Lines 357-407: Several text corrections.*

Text has been amended following the reviewer recommendations. Some parts have been rewritten to improve readability.

**Discussion**

**Structural and stratigraphic meaning of the Lower Parautochthon synorogenic basins**

*Lines 411-494: Several text corrections.*

Text has been amended following the reviewer recommendations. Some parts have been rewritten to improve readability.

**Source-areas of the siliciclastic rocks and olistoliths in the Lower Parautochthon: MDS results**

*Line 495 (sub-section title): Change "Source-areas" by "Provenance" and delete "MDS results".*

We have changed the title according to the reviewer comment.

*Lines 497-552: Several text corrections.*

Text has been amended following the reviewer recommendations. Some parts have been rewritten to improve readability.

**Origin of Variscan Zircons**

*Line 353 (sub-section title): Change "Variscan zircons" by "Variscan detrital zircon grains".*

We have changed the title according to the reviewer comment.

*Lines 544-580: Several text corrections.*

Text has been amended following the reviewer recommendations. Some parts have been rewritten to improve readability.

*Lines 581-582: Note that in SW Iberia there are Late Devonian volcanic rocks (i.e. Cercal porphyries).*

This has been noticed. However we have change our discussion and changed figure 14 accordingly, as we do not see evidences for far traveled "airborne" zircons, carried in volcanic ash clouds.

**Conclusions**

*Lines 587-618: Several text corrections.*

Text has been amended following the reviewer recommendations. Some parts have been rewritten to improve readability.

**General comment:** *I would like to see more explored the following topic: The provenance of Silurian-Mid-Ordovician zircon grains found in the GTMZ Lower Parautochthonous units. They derived directly from magmatic rocks of Gondwana? or from Laurussia? If they derive from Laurussia, what Paleozoic terrain will they originate from? Meguma, West Avalonia, Ganderia, East Avalonia?*

We have made some comments in the discussion in respect to the source of the main zircon age peaks in the synorogenic basin of NW Iberia. According to field and geochronological data of the surrounding domains (allochthonous and autochthon), all these age groups are represented in nearby sources. Also, to date there is no data supporting the presence of far sources in this basin.

**R2 – John Wakabayashi**

**Introduction**

*Lines 41-107: Several small text corrections.*

Text has been amended following the reviewer recommendations. Some parts have been rewritten to improve readability.

**Geological Setting**

Text has been amended following the reviewer's recommendations. Some parts have been rewritten to improve readability.

**Review of the synorogenic marine sequences in NW Iberia**

*Lines 167-192: Small text corrections.*

Text has been amended following the reviewer recommendations. Some parts have been rewritten to improve readability.

*Line 193: Earlier in the paper, the use of the term "exotic" should be defined. What is meant by a lithified grain? I would presume that this means lithic grain instead of a detrital mineral grain. I would also guess that "exotic" is an adjective acting on "lithic" in which the statement should be "exotic lithic" without "and". "And" infers that exotic and lithic grains may be two distinct categories.*

We have clarified this issue earlier in the text. We added text to explain the difference between exotic (extrabasin) and native (intrabasin) clasts, including lithified (consolidated) pebbles and unconsolidated sediments (intraclasts).

*Lines 195-233: Several text corrections.*

Text has been amended following the reviewer recommendations. Some parts have been rewritten to improve readability.

*Line 234: "Lydite" and "ampelite" are not terms commonly used in published papers in English language journals. In fact this is the first time I have seen those terms. In looking up the terms,*

*"Lydite" should be replaced by "radiolarite" or "radiolarian chert" and I'm not exactly sure what should replace ampelite, which is supposed to be some sort of shale/slate/phyllite/schist rich in carbonaceous matter and possibly pyrite.*

We kept the original terminology because lydites correspond to black cherts and ampelites to graphitic slates. Text has been added to fix this problem, explaining both terms (of general use in NW Iberia bibliography).

**Extending the Lower Parautochthon**

*Lines 243-250: Minor text corrections.*

Text has been amended following the reviewer recommendations. Some parts have been rewritten to improve readability.

**Stratigraphic sequences and youngest zircon ages**

*Lines 252-262: Minor text corrections.*

Text has been amended following the reviewer recommendations. Some parts have been rewritten to improve readability.

*Line 263: Because U-Pb detrital zircon geochronology is a key component of this paper there needs to be more given on methods: not only the analytical methods, but, also the data analysis. The should include, but not be limited to, how maximum depositional ages are defined given that different groups do this differently.*

Text has been amended following the reviewer recommendations. The complete description of the analytical and data analysis methods is in Supplementary File.

*Line 265: define "YZ". I presume this means young zircon population. How does this differ from the "maximum depositional age"*

Text has been added following the reviewer recommendations.

*Lines 266-315: Minor text corrections.*

Text has been amended following the reviewer recommendations. Some parts have been rewritten to improve readability.

**Lower Parautochthon magmatic olistoliths, their ages and possible source-areas**

*Lines 317-328: Several small text corrections.*

Text has been amended following the reviewer recommendations. Some parts have been rewritten to improve readability.

**Magmatic olistoliths age results**

*Line 331: Are these different than the four samples noted in the Sec. 5 above. I suspect they are the same, but if so, this lead sentence should be rewritten, so as not to restate what was already written.*

Text has been amended following the reviewer recommendations. Some parts have been rewritten to improve readability.

*Line 333: foliated implies metamorphic. If so, detail metamorphic as well as original igneous mineralogy.*

Text has been amended following the reviewer recommendations.

*Line 338: See previous sample-if metamorphosed, metamorphic mineralogy should be explained.*

Done, for all the metamorphic olistoliths described in this section.

*Line 348: If the block-matrix textures indicate that this is an olistostrome then this is fine, but an older age in younger matrix can be emplaced by tectonism by "plucking" of material far downdip-at least this is the explanation put forth by advocates of return-flow mélanges (but certainly not supported by all mélange researchers). However, the stratal disruption of an originally syn-sedimentary volcanic flow is precluded; see below*

The text has been rewritten to improve readability.

*Line 353: See above.*

As in the previous case, the text has been rewritten to improve readability.

**Olistoliths magmatic ages from references**

*Lines 317-328: Small text corrections.*

Text has been amended. Some parts have been rewritten to improve readability.

**The possible sources of the magmatic olistoliths are in the UPa**

Text has been amended following the reviewer 1 recommendations. Some parts have been rewritten to improve readability.

**Discussion**

**Structural and stratigraphic meaning of the Lower Parautochthon synorogenic basins**

Text has been amended following the reviewer 1 recommendations. Some parts have been rewritten to improve readability.

**Source-areas of the siliciclastic rocks and olistoliths in the Lower Parautochthon: MDS results**

Text has been amended following the reviewer 1 recommendations. Some parts have been rewritten to improve readability.

**Origin of Variscan Zircons**

Text has been amended following the reviewer 1 recommendations. Some parts have been rewritten to improve readability.

*Line 584: In an orogenic environment long distance transport from source to depocenters is not exceptional. The on-land part of the sediment transport systems can include large rivers that transport detritus (including zircons) long distances and not always "normal" to the trend of the orogen. The same can be said of submarine sediment movement that can commonly show a large along-strike component of transport (currents parallel to long axis of basin). Good examples of this have been presented for western North America by Dumitru et al (2013 Geology 41, 187-190) and Dumitru et al. 2016 (Geology 44, 75-78)*

We have rewritten this sub-section. We have reconsidered some of the previous possibilities, and we found that it is not necessary to explain far-traveled zircon grains with the data we have. It is easier, and more elegant, to explain all zircon populations of the synorogenic basin coming from in nearby sources, found in the autochthonous and allochtonous domains.

**Conclusions**

No comments. Text was improved to increase readability.

**General comments:**

1) *Although describing block-in-matrix units is not the main goal of this paper, the field and petrographic observations are important because they are relevant to global debates on mélange origins. For this reason, I request that the authors add a bit more to their map scale, outcrop scale, and petrographic scale observations. Owing to the detail and quality of the existing observations, I suspect these additions can be made easily.*

Yes, you are right, but this is not within the main scope of this work. The manusctipt is already thick enough with so many data. We expect to write a work dedicated to the detailed description of the facies and the nature of the olistoliths in the NW Iberia synorogenic marine basin.

2) *The types of observations I recommend are those that show the similarities and differences between tectonic mélanges and olistostromes. For example, the metamorphic assemblages in the rocks (if metamorphosed) should be described. Based on the existing descriptions it seems to me that some of the olistostromal blocks in the study area are higher in metamorphic grade than the matrix and that there is a range of metamorphic grade encompassed by the blocks in the olistostrome. Specific details should be given. In contrast it seems to me that the tectonic mélanges in the study area*

*do not have blocks of higher metamorphic grade than the matrix or flanking units and there all of the blocks are isofacial, but the details are not given in the paper: they should be. These details are a subset of the larger relationship: exotic versus native blocks. The reason why I mentioned metamorphism, is that it can be difficult to tell if a block is exotic whereas, a block of higher metamorphic grade than flanking units or matrix is clearly so. In the paper it seems as if exotic blocks are confined to the olistostromes, whereas the tectonic mélange contains only blocks derived from the flanking units (or the specific disrupted zone) so that the blocks are entirely native. This should be clarified in the paper.*

Because we also think that these descriptions can be important to this manuscript, we have added text dedicated to (briefly) describe some of the sedimentary, metamorphic, and tectonic textures in the flysch sequences and exotic clasts and olistoliths. As in the previous point, we will soon produce a manuscript dedicated to the study of the internal structures of the BIMF deposits and their relation with the basement and tectonically overriding units.

**3)** *It seems to me that authors assume that readers will see the relationships summarized above as being clearly connected to origins as olistostromes or tectonic mélanges, hence they do not see the need to expand further on their observations. Yet there are many researchers who assert that the presence of exotic blocks is evidence for tectonic incorporation of blocks into matrix.*

As in the previous points, you are right. We know about this problem and we hope that with the new writing this is clearer for the reader.

**4)** *This brings up a still broader issue/concept, which is the definition of tectonic versus sedimentary versus "polygenetic" mélange. These terms are connected with the primary mode of mixing of blocks into matrix. In an olistostrome (sedimentary mélange), the blocks are mixed by sedimentary process, whereas a tectonic mélange, mixing of blocks into matrix (and creation of blocks) is a result of tectonic strain. For most readers, the meaning of "polygenetic" is not so easy to grasp in the sense of block/matrix relationships. It seems to me that many people mistakenly believe that this is simply the imposition of deformation on a sedimentary mélange but "polygenetic" means that additional blocks are created by deformation within and on the flanks of a sedimentary mélange: this can happen because of the creation of tectonic block creation by faulting of intact units bordering the sedimentary mélange and/or as a result of tectonic strain fragmenting some olistostromal blocks. A short explanation of the definitions of sedimentary, tectonic, and polygenetic mélanges should be given early in the paper, rather than simply referring to the definitions given in the cited papers.*

We have added a simple definition of the three kinds of mélanges found in this sector.

**Author's changes in manuscript (and supplements)**

Moderate changes have been applied to the manuscript, using the comments and revisions of Manuel Francisco Pereira (MFP) and John Wakabayashi (JW). We have accepted most of the proposed changes, we have improved English writing and added new text to avoid repetitions. We have introduced some general concepts and synthesized regional geological aspects common to all stratigraphic units of the Lower Parautochthon in NW Iberia. We have also improved the original version of the Supplementary File, to follow with the changes made in the manuscript.

All sections, including the manuscript title, have text changes (see manuscript with track changes on). Most changes are formal and grammatical aspects that were improved. However, some sections suffered some reorganization, because of text travelling and integration of concepts.

As MFP suggested, figure 4 (sample map) and table 1 have migrated to the Supplementary File (SF). Table 1 is named as table SF1, and figure 4 is now the revised and updated figure SF1.1. The SF figures were all relabeled.

We have made two new figures (4 and 5), showing the regional tectono-stratigraphic correlation of the Lower Parautochthon synorogenic units, the position and distribution of the zircon ages, nature of the olistoliths, main sedimentary and tectonic structures, and fossiliferous content. Figures 5-13 have been renumbered according to the new figure scheme. Figures 10, 12 and 13 (new figures 11, 13 and 14) were updated following MFP comments and other changes we made in section 6.3 of the manuscript. Figures 2, 3 and 7 were also updated, matching to the changes made.

For more information, see the revised manuscript with the active track changes, attached below in this document.

Once again, we had sincerely valued the work of the reviewers and the editor whose comments helped to improve majorly the quality of the manuscript. We hope that it can be now accepted for production in its current version.

On behalf of the authors of this manuscript, Thank you very much!

Kindest regards.

*Ícaro Fróis Dias da Silva* (corresponding author)

[revised manuscript text omitted]
 stratigraphiestratigraphy and deformative aspectsstructural features of the lowerlowermost tectonic sheets of Pa in the transition toPa, directly above the CIZ, pointed to that section was part ofwere interpreted as a synorogenic basin with possible (Middle-Late) Devonian age (Antona and Martínez Catalán, 1990; González Clavijo and Martínez Catalán, 2002; Martínez Catalán et al., 2004; Pereira et al., 2009; Rodrigues et al., 2013).

ThisThe attempt to better understand the tectonostratigraphy of the Pa, led to a later division in two tectonically overridingstacked units, Upper and Lower Parautochthon (UPa and LPa), in the meaningsense firstly proposed by Rodrigues et al. (2006a; 2006b2006; 2013) and updated by Dias da Silva et al. (2014; 2015; 2016). This division restrictslimits the UPa to a pre-Variscan upper Cambrian-Silurian sequence comparable with the CIZ and LA that was affected by Variscan recumbent folds and thrusts; and defines the LPa as an imbricated thrust sequence bearing slices of a foreland synorogenic basin, with the

165  younger slices in the transition to the CIZ (e.g. Martínez Catalán et al., 2016). This proposal required a relocation of the bounding The thrust fault structures ofbounding the lower tectonic sheets of the GTMZ: are (Figs. 2 and 3):

 i) the LA UPa thrust ( Basal Thrust (LABT, Figs. 3 and 4) or basal thrust of the Centro-Transmontano thrust complex (in the meaning of Ribeiro et al. 1990b) into an upper structural position1990) represents the roofing thrust of the Parautochthon;

170 ii) the UPa-LPa thrust system named as (Main Trás-os-Montes Thrust (, MTMT) (; Ribeiro, 1974; Ribeiro and Ribeiro, 2004; Meireles et al., 2006; Pereira et al., 2006) to a lower structural position (Dias da Silva et al., 2014). The MTMT is an up to 1000 m thick gently dipping low-grade shear-zone responsibleinterpreted to be caused by the thrusting of the upper Cambrian-Silurian (pre-orogenicpreorogenic UPa) sequence onto the syn-orogenicsynorogenic LPa, allowing theproducing significant crustal thickening the upper crust during

175 the Tournaisian-Visean stage (Dias da Silva et al., 2014; 2015; 2016; 2020; Azor et al., 2019);

 iii) At the base of the LPa another major bedding-parallel fault structure named the Basal Lower Parautochthon Detachment (BLPD), also gently dipping, separatesis the syn-orogenic piggy-backsole fault separating the synorogenic imbricated slices from the structurally underlying non imbricated autochthon (Dias da Silva et al., 2014). The BLPD was developed followingusing a slip-favorable stratigraphic unit, the condensed

180 Silurian autochthonous sequence formed by Silurian carbonaceous cherts and graphitic-siliceous slates (SCSS) (González Clavijo and Martínez Catalán, 2002; Dias da Silva et al., 2014).

‑ ‑ ‑ ‑ ‑ ‑ ‑ ‑ ‑ ‑ ‑ ‑ ‑ ‑ ‑ ‑ ‑ ‑ **Con formato:** Fuente de párrafo predeter., Fuente: Times New Roman, Inglés (Reino Unido)

As theThe northern CIZ autochthonous domain presentsconsist of an Ediacaran to Lower Devonian preorogenic sequence, disturbed by two regional unconformities, and including c. 490 Ma to 460 Ma felsic to intermediate, locally mafic magmatism, Floian Armorican-type quartzites, a Middle-Upper Ordovician mostly detrital sequence and the SCSS (e.g. Sousa, 1984;

185  Valladares et al., 2000; GutierrezGutiérrez- Marco et al., 2019; Sánchez García et al., 2019),). In the autochthon laying immediately below the BLPD separates the LPa without substantial upper crustal thickening, which, tectonic overburden mainly occursoccurred due to the piggy-backaction of thin-skinned imbricated thrust-duplexes rooted in the BLPD, developed in the syn-orogenicLPa tectonic units (Fig. 2 and sections 4 and 5 in Fig. 33) (Dias da Silva et al., 2020). In the sectors where the LPa is present, thickening in both CIZ and LPa was rapidly attenuated by the succeeding synorogenic extensional processes

190  (Dias da Silva et al., 2020).

‑ ‑ ‑ ‑ ‑ ‑ ‑ ‑ ‑ ‑ ‑ ‑ ‑ ‑ ‑ ‑ **Con formato:** Fuente de párrafo predeter., Fuente: Times New Roman, Inglés (Reino Unido)

In the studied area, the three structural units considered in this work (UPa, LPa and CIZ) underwent a regional Barrovian metamorphism (M₁) through the early Variscan compressive events (C₁+C₂ on the AlcockMartínez-Catalán et al., 20152014 proposal; c. 360-330 Ma) which were later followed by a complex extensional (E₁-M₂; c. 340-320 Ma) and compressivecompressional (C₃-M₃; c. 318-300 Ma) tectonothermal history (Dallmeyer et al. 1997, Azor et al., 2019; Dias da

195  Silva et al., 2020). Some specialized studies were performed2020, and references therein).

The LPa synorogenic ensemble was also affected by metamorphism of very low- to low-grade (chlorite zone). Attempts to discriminate if the metamorphic grade at the syn-orogenic units was lower in the synorogenic units than in the preorogenicpreorogenic units byusing illite crystallinity (Antona and Martínez Catalán, 1990) and Colour Alteration Index in conodonts (Sarmiento and García-López, 1996; Sarmiento et al., 1997) but no conclusive results were attained. This matches with theinconclusive. Petrographic observations made by Matte (1968) conclusions forin the autochthonoussynorogenic deposits of the San Clodio syn-orogenic deposits, supporting the same metamorphism and deformation in the Carboniferous than in the CIZ to the SE of Monforte (Fig. 2) and in the underlying autochthonous Ordovician sequenceOrdovician sequence, and by Dias da Silva et al. (2020) in the LPa and CIZ in the eastern rim of Morais Complex, shows similar low grade epizone metamorphism in both pre- and synorogenic sequences. However, the San Clodio flysch rests unconformably above the reverse limb of a large $C_1$ recumbent syncline whose axial planar cleavage is more evolved than that of the flysch above (Martínez Catalán et al., 2016). And it is also older: c. 360 Ma (Dallmeyer et al., 1997), while detrital zircons are as young as 324 Ma in the San Clodio Series and 340 Ma in the synorogenic deposits of Trás-os-Montes (Martínez Catalán et al., 2004, 2008, 2016), age of emplacement of the Allochthonous Complexes during $C_2$. The first foliation in the preorogenic metasediments of the UPa and CIZ is axial planar to recumbent folds of the $C_1$ event, and predates the main foliation in the synorogenic deposits. But a second, low grade penetrative foliation was developed in the UPa, LPa and CIZ during the emplacement of the Allochthon ($C_2$). This second regional foliation is the one showing similar aspect and metamorphic conditions in both UPa and LPa ensembles, but in the latter it represents the first tectonic fabric.

**3 Review of the syn-orogenicsynorogenic marine sequences in NW Iberia variscan hinterland**

The internal zones of the orogenic belts are considered areas with scarcely preserved related syn-orogenicsynorogenic sequences because of the followingsubsequent denudation caused forby the orogeny itselforogenic relief (Martínez Catalán et al., 2008). Nevertheless), but they might be preserved in the core of synclines or below post-depositional thrust. In other parts of the Variscan belt, Franke and Engel (1986) stated the existence ofdescribed tectonic slices carrying syn-orogenicsynorogenic sedimentary units from the more internal areas in several European Variscan massifs.. In NW Iberia, while there is a complete record of the synorogenic deposits in the foreland fold and thrust belt (CZ. e.g.a Marcos and Pulgar, 1984; Merino-Tomé et

[revised manuscript text omitted]

The structurally underlying Rábano Fm. (González Clavijo and Martínez Catalán, 2002), structurally underlying also called
External Gimonde, in Martínez-Catalán et al. (2016), comprises diverse lithologies being the most generalabundant a block-
in-matrixBIMF sequence displaying profuse majorlarge exotic and lithified olistoliths of rhyolite, acidic volcanic
tuffsdeformed rhyolites and dacites, felsic metatuffs, epiclastic volcanic rocks, white and graygrey quartzite, blackSilurian
lydite (black radiolarite) and ampelite. (carbonaceous shale), greywacke, phyllite, and limestone (Fig. 5A6A). The ages of
these big blocks (sometimes hundreds of metres in length) based on radiometric agesfossils and fossiliferous content rangesU-
Pb zircon ages range from Furongian to Emsian (González Clavijo et al., 2012; 2016). At the upperuppermost Rábano Fm. a
flyschoid sequence made of phyllite, quartzlitharenite, and local polygenic microconglomerate holds exotic, lithified and
deformed grains andexotic clasts and lithified intraclasts (Fig. 6A7A, B, and C); including plagioclase and volcanic quartz
mineraloclasts and explosive quartz shards supportingindicating a vulcanite richnearby volcanic source area for this unit
(González Clavijo, 2006). Detrital zircon ages studies performed in this wild-flysch (sample SO-9; situation in Fig. 46) support
a syn-orogenicsynorogenic nature and points to an age Tournaisian or youngera Visean MDA (González Clavijo et al., 2012;
2016; Martínez Catalán et al., 2016). The Rábano Fm. lies on a sheared condensed black Silurian unit which develops a tectonic
mélange and, sometimes, a polygenetic mélange where deformation is superimposed to a sedimentary mélange. ; Martínez
Catalán et al., 2016).

The discontinuity of the Silurian unit along the tectonic limit suggests that a strong stretching event was concentrated in this
band.

Next unit is formed by the San Vitero flyschFm. (Martínez García, 1972) is a flysch made of up to metre thick phyllite and
quartzlitharenite rhythms up to metre thick and local lenses of microconglomerate withpolygenic microconglomerates holding
exotic, clasts and lithified and deformed grainsintraclasts (Figs. 7D, 7E, 8A and pebbles (Fig. 6D and E8B) (
[revised manuscript text omitted]
 andit belongs to a horse with the sheared black Silurian unitSCSS at the base (Fig. 7A). The8A). This sample yields a magmatic concordia age of 476.0 ± 1.5 Ma age (Floian) (SI-9B); but the). Its
515 position of the block,within an olistostrome stratigraphically higher than the Silurian black unit dated by graptolites (González Clavijo, 2006, pers. com. Gutiérrez Marco, Sá, and Piçarra on this locality),graptolite-rich SCSS (Fig. 8A) excludes other plausible explanations as an interlayered pyroclastic flow, or a sill. (González Clavijo, 2006; Gutiérrez Marco, Sá, and Piçarra pers. com.).

520 **5.2 Olistoliths magmatic Published ages from referencesmagmatic olistoliths**

In the Alcañices synform a quantity of vulcanite several olistoliths hasof felsic volcanic rocks have been identified; all of them of rhyolite to dacite composition, and often forming biglarge clusters with aelongated NW-SE attitude.

Previous research (González Clavijo et al., 2016) obtained an age of the Nuez olistolith (NUEZ-01; situation; see location of all samples in Fig. 4the Supplementary File), one of the major blocks forming a several kilometer kilometers-long cluster
525 included in an olistostrome inside the syn-orogenicsynorogenic Rábano Fm., towards the Ssouthern limb of the Alcañices synform. This block contains two volcanic facies: dacite lava and dacitic pyroclastiequartz-eyed tuff (Ancochea et al. 1988), belonging the attained age to the latter. A 1988). The LA-ICP-MS U-Pb in zircon concordantisotope analysis of magmatic

zircons of a dacitic tuff sample returned a concordant magmatic age of 497 ± 2 Ma (lowermost Furongian) was achieved, which was considered magmatic.).

530 In the Nnorthern limb of the same synform other volcanic body, , the Figueruela dacite (COS-8; situation in Fig. 4), was dated by SHRIMP-II U-Pb analysis (Farias et al., 2014) and yielding a magmatic concordia age of 488.7 ± 3.7 Ma (around the limit Furongian/Tremadocian). This igneous rock was interpreted as a flow of dacitic lava flow interlayered in the Paraño Group of the Galicia Schistose Domain or Parautochthon *sensu lato*. A thoroughOur field review ofwork in the area shows a has revealed that the Figueruela dacite belongs to a major cluster of olistoliths in a large mass wasting deposit, mainly composed of blocks

535 of felsic lavas (dacite and rhyolite lavas) and tuffs, but also containing large quartzite and black lydite big blockslenses. In our reinterpretation the Figueruela dacite is an olistolith contained in a basal block-in-matrix unit placed below the San Vitero Fm. coherent primary unit and above the black Silurian condensed unit. In this section, parts of the block-in-matrix unit and the Silurian black sequence are sheared, forming a polygenetic mélange in the meaning proposed by Festa et al. (2019, 2020). SCSS. Thus, in our general model,here we support that the Figueruela dacite belongs to the syn-orogenicsynorogenic LPa as

540 proposedpreviously stated by González Clavijo et al. (2016). The SHRIMP U/Pb in zircon age is 488.7 ± 3.7 Ma (around the limit Furongian/Tremadocian) and is considered magmatic.
Between the Alcañices and Verín synforms, at the NAt the northern edge of the Bragança Allochthonous Complex, other bigsignificant volcanic body was dated by Farias et al. (2014) and named as, the Soutelo rhyolite (COS-7; situation in Fig. 4). This sample), was dated by the same analytical method than the Figueruela daciteSHRIMP-II U-Pb analysis (Farias et al.,

545 2014) yielding a 499.8 ± 3.7 Ma (upper Miaolingian). This massive) concordia age. The aforementioned authors have included the Soutelo rhyolitic lava was also included in the so-called Paraño groupGroup and considered ait as volcanic event amongin the preorogenic sedimentary sequence. of the (Upper) Parautochthon. Our field study disclosed the existence of a hugean important cluster of olistoliths of diverse lithologies as black lydite, grey quartzite, greywacke, limestone, rhyolitic lavas, and acidic pyroclastic tuffs, being the last two the most abundant types. Complementarily, this major block-in-matrix unit is placed

550 on top of an intensely deformed condensed black Silurian unit (a tectonic mélange).the mylonitized SCSS that defines the BLPD. For all these reasons we consider that the Soutelo rhyolite volcanic body analysed in Soutelo is also an olistolith inside the syn-orogenicsynorogenic LPa.

**5.3 The possible sources of the magmatic olistoliths are in the UPa**

555 The UPa unit, structurally below the Lower Allochthon as defined forby Dias da Silva et al. (2014) belowin the eastern rim of the Morais Complex, (Figs. 2 and 3), contains an Uppera late Cambrian to Silurian detrital sequence with minor limestones and interbedded voluminous volcanism (Pereira et al., 2000; 2006). The main volcanic events are, from bottom to top, the Mora acid and basic volcanites (felsic to mafic volcanic rocks (Mora Volcanics; Dias da Silva, 2014; Dias da Silva et al., 2014; Díez-Montes et al., 2015); the traditionally named asSaldanha gneiss (Ribeiro, 1974; Ribeiro and Ribeiro, 2004; Pereira et

560 laal., 2006; Pereira et al., 2008) which actually is a rhyolitic dome composed by lavas and volcanic tuffs (Dias da Silva et al.,

2014; Díez-Montes et al., 2015);and a large felsic and finally the big acid and basicmafic volcanic half of the sedimentary complex (
[revised manuscript text omitted]
 preferably in black Silurian rocksthe SCSS (here inferredinterpreted as the LPBDBLPD). Immediately above the LPBDBLPD, a low metamorphic grade detritalturbiditic sequence is exposed at the coast linecoastline, where the PicónPICON-2 sample was collected (Figs. 2, 3, 4, 5 and 4) and the

detrital zircon study supports theSupplementary File) giving support for a Variscan syn-orogenicsynorogenic origin of this sequence (see above) and consequently we ascribe itthe ascription of the lower part of the Rio Baio thrust sheet to the LPa.

695 Thus at the base of, under the Cabo Ortegal Complex, in the Rio Baio tectonic unit, the UPa/LPa division is also present, with a thin LPa structural unit placed ontoParautochthon comparable with that of the Morais and Bragança areas exists, with a little deformed thin LPa unit overlying a tectonic mélange developed in the Silurian rocks; while SCSS, and the restupper part of the Rio Baio thrust sheet is here proposed asabove, representing the pre-orogenicpreorogenic UPa unit in the Cabo Ortegal Complex on base of lithological correlation, metamorphic grade and deformation. An isotopic age attainedobtained by

700 Valverde Vaquero et al. (2005) in the Queiroga alkaline rhyolite (U-Pb TIMS, 475±2 Ma - Tremadocian) reinforces Floian) supports this ascription for similarityby comparison with other acidic volcanites infelsic volcanic rocks of the UPa around the Morais Complex UPa (Dias da Silva, 2014; Dias da Silva et al., 2014; Dias da Silva et al., 2016; this work data).

Based on the geochronology results of the 17 magmatic and detrital rock samples here presented plus previous research here and previously published zircon age data we propose interpret that the LPa Variscan syn-orogenicsynorogenic sedimentary

705 and structural unit is general in the NW Iberia, forming a forms a continuous tectonic carpet which separatesunderlying the GTMZ and separating it from the CIZ. Nevertheless, as can be seen in Figs. 2 and 3, theThe LPa is not observed in some reaches of the zones limit for different reasons. In some parts between the two zones because if existing, it has been hidden by late Variscan transcurrent faults or due to the intrusion of Variscan granitoids have intruded, erasing the previous geological information.. Between the Cabo Ortegal and the Bragança complexes and in the northern Porto sector the available data from

710 referencesthe literature does not conclusively support the existence of syn-orogenicsynorogenic sequences which could be endorsed to the LPa;, and no detrital zircon study oriented to this targetstudies have been performed yet, being a future aim of the research team. Finally, it is worth to highlight the existence of a detached remnant of LPa sequences (until now. Only the San Clodio series) preserved at the core of a late Variscan syncline in the autochthonous side of the limit (CIZ) Series may represent a link between the LPa of Cabo Ortegal and Trás-os-Montes, although it is younger and, although imbricated, it is

715 not far away from the LPa/CIZ boundaryfully allochthon.

The strongly deformed black Silurian condensed sequence SCSS present at the base of the every LPa tectonic slice, and frequently separated from the syn-orogenicsynorogenic sequence forby a thrust fault rooted in the BLPD, must be considered a tectonic mélange in the meaning proposed by Festa et al. (2019; 2020) as it also incorporates tectonic blocks and olistoliths from the base of the syn-orogenicsynorogenic sequences; so. So, when the shearing bandrelated shear zone incorporates glided

[revised manuscript text omitted]

**Section 1**

W     E

ORTEGAL COMPLEX

MTMT   LPBD

Viveiro Fault

Present erosional surface

Ollo de Sapo

**Section 2**

SW     N

SIL SYNCLINE

Ollo de Sapo Anticlinorium

Piornal Anticline

Courel Syncline

N S   LPBD

Ollo de Sapo

**Section 3**

SSW     NNE

MORAIS COMPLEX

BRAGANÇA COMPLEX

Chandoiro Fault

LPBD MTMT

Ollo de Sapo

**Section 4**

SSW     NNE

MORAIS COMPLEX

Villadepera Antiform

ALCAÑICES SYNFORM

LPBD MTMT

MTMT

LPBD

Miranda do Douro Orthogneiss

Ollo de Sapo

**Section 5**

SSW     NNE

ALCAÑICES SYNFORM

LPBD

Ollo de Sapo

THRUST FAULTS

DETACHMENT FAULTS

LATE EXTENSIONAL FAULTS

STRIKE-SLIP FAULTS

OTHER FAULTS

MTMT    MAIN TRÁS-OS-MONTES THRUST

LPBD    LOWER PARAUTOCHTHON BASAL DETACHMENT

**Section 6**

W     E

MORAIS COMPLEX

MTMT LPBD

MTMT

LPBD

[revised manuscript text omitted]

Peso Fm. volcanics
P-385
P-381
LPa volcanic olistoliths
EC-PO-419
EC-PO-337
VC-21ZIR
SO-6
MEI-ZR-01
CR-ZR-01

Age (Ma)

*NW Iberia synorogenic basins and possible source-areas*

Reference sources
Volcanic rocks
Siliciclastic rocks

**Olistoliths**

Armorican Quartzite with skolithos
Middle-Upper Ordovician volcanic rocks
Greywackes and other siliciclastic deformed rocks

*Probability density plot areas*

Variscan ages
Ediacaran-Paleozoic ages
Scattered "old" ages
Mesoproterozoic and Tonian "gap"
Paleoproterozoic and Archean ages

**CR-ZR-01** New U-Pb zircon ages
**SO-6** U-Pb zircon ages from references
**LA-1** Reference sources
**SO-11** Excluded from the study
(n<25 concordant ages; inc. sources)

[Figure]

**Figure 12: Age distribution plots of the Cluster 1 type populations: "Upper Parautochthon Middle Ordovician-Silurian volcanism". See text and Supplementary File for a detailed description.**

[Figure]

[Figure]

**Figure 1213: Age distribution plots of all groups belonging to the Cluster 2 populations: "Multiple Gondwana-derived sources".**
**Legend is in Fig. 1112. See text and Supplementary File for a detailed description.**

[Figure]

[Figure]

**Figure 14: Sketch of the orogenic collision at the Tournaisian-Visean displaying the trench-fill turbidites and the block-in-matrix deposits. The upper part represents the input of the zircon populations from different sources and zones.**

1580

**Table 1: UTM coordinates and short description of the 17 samples gathered in this work.**

| SAMPLE ID | UTM system | Zone | Easting | Northing | Short description |
|---|---|---|---|---|---|
| **DETRITAL** | | | | | |
| PICÓN-2 | WGS84 | 29T | 601064 | 4844313 | Fine-grained black quartzites with pyrite and black pelite laminations |
| EC-PO-293 | WGS84 | 29T | 675598 | 4642546 | Quartz lithic sandstone (Gimonde type); impure grey quartzites <10cm clasts. |
| GIM-ZR-01 | WGS84 | 29T | 674114 | 4644518 | Lithic microconglometate, lithic-sandstones and phyllite rhythms. Light brown colour. |
| AD-PO-66 | WGS84 | 29T | 673233 | 4643613 | Quartz lithic sandstone and phyllite rhythm. Feldspar and quartz eyes: pyroclastic? |
| AD-PO-48B | WGS84 | 29T | 621227 | 4596860 | Massive fine grain sanstone. |
| AD-PO-57 | WGS84 | 29T | 642881 | 4597741 | Quartz lithic sandstone and phyllite rhythm (cm-dm). |
| AD-PO-55 | WGS84 | 29T | 644424 | 4595258 | Quartz lithic sandstone and phyllite rhythms. Fining upwards. |
| AD-PO-49 | WGS84 | 29T | 645744 | 4594231 | Quartz lithic sandstone and phyllite rhythms (cm-dm). |
| MIR-41 | WGS84 | 29T | 645735 | 4594230 | Quartz lithic sandstone and phyllite rhythmss (cm-dm). |
| MEI-ZR-01 | WGS84 | 29T | 683703 | 4569953 | Middle-grained greyish lithic sandstones with sedimentary laminations. |
| CR-ZR-01 | WGS84 | 29T | 678159 | 4569706 | Greenish-grey fine grained massive quartz lithic sandstone. |
| **OLISTOLITHS** | | | | | |
| EC-PO-337 | WGS84 | 29T | 692792 | 4642324 | Rhyodacitic pyroclastic tuff, quartz eyes and shards. Inside quartz lithic sandstone. |
| EC-PO-419 | WGS84 | 29T | 663291 | 4643585 | Acidic pyroclastic tuff, quartz eyes and S2 pervasive foliation. |
| PET-01 | WGS84 | 29T | 706355 | 4638642 | Massive reddish rhyolite. Py crystals. Grey when fresh. |
| RAB-01 | WGS84 | 29T | 730014 | 4628291 | Whittish rhyolitic pyrockastic tuff. Quartz eyes and shards. Inside grey fine sandstone. |
| **UPPER PARAUT** | | | | | |
| P-381 | WGS84 | 29T | 674426 | 4603329 | Intensely foliated white rhyolite with quartz phenocrysts |
| P-385 | WGS84 | 29T | 681845 | 4609510 | Foliated dacites with quartz, feldspar and plagioclase phenocrysts |